



# Measurement Report: Ten-year trend of PM2.5 major components and source tracers from 2008 to 2017 in an urban site of Hong Kong, China

Wing Sze Chow[1], Kezheng Liao[1], X. H. Hilda Huang[2], Ka Fung Leung[2], Alexis K. H. Lau[2], Jian Zhen Yu[1,2]

[1]Department of Chemistry, The Hong Kong University of Science and Technology, Clear Water Bay, Kowloon, Hong Kong, China

[2]Division of Environment & Sustainability, The Hong Kong University of Science and Technology, Clear Water Bay, Kowloon, Hong Kong, China

*Correspondence to*: Jian Zhen Yu (jian.yu@ust.hk)

**Abstract.** Fine particulate matter (PM2.5) remains a major air pollutant of significant public health concerns in urban areas. Long-term monitoring data of PM2.5 chemical composition and source-specific tracers provide essential information for identification of major sources as well as evaluation and planning of control measures. In this study, we present and analyze a ten-year data set of PM2.5 major components and source-specific tracers (e.g., levoglucosan, hopanes, $K^+$, Ni, V, Al, Si) collected over the period of 2008-2017 in an urban site in Hong Kong, China. The time series of pollutants were analyzed by

the Seasonal and Trend decomposition by Loess method and general least squares with Autoregressive-Moving average method. Bulk PM2.5 and all its major components displayed significant decline of varying degrees over the decade. PM2.5 was reduced by 40% and at -1.5 µg m$^{-3}$yr$^{-1}$. PM2.5 components that are predominantly influenced by local vehicular emissions showed the steepest decline, with nitrate by -84%, elemental carbon by -56%, and hopanes by -66%, confirming effective control of local vehicular emissions. For components that are significantly impacted by regional transport and secondary

formation, they had a notably lower percentage reduction, with sulfate by -33% and organic carbon by -23%, reflecting complexity in their region-wide contributing sources and formation chemistry. Levoglucosan and $K^+$, two tracers for biomass burning, differed in their reduction extent, with $K^+$ at -60% and levoglucosan at -47%, indicating they likely track different biomass burning types. Dust components in PM2.5 also decreased, by -37% for Al and -46% for Si. The year of 2011 was an anomaly in the overall trend in having higher concentrations of PM2.5 and components than its adjacent years, and the long

time series analysis attributed the anomaly to unusually lower rainfall associated with strong La Niña events. This ten-year trend analysis based on measurements exemplifies the utility of chemical composition data in support of an evidence-based approach for control policy formulation.

## 1 Introduction

Air pollution controls are of both local and global importance. Their effectiveness needs to be periodically reviewed for

optimizing options to improve air quality and minimize environmental impacts. Particulate matter with aerodynamic diameter less than 2.5 µm, namely fine particulate matter (PM2.5), is a major air pollutant. It is a significant contributor to visibility reduction, climate change, and detrimental effects on human health (Yang et al., 2018; Zhao et al., 2013; Lippmann and Chen, 2009; Ko et al., 2007; Kim et al., 2006; Cheung et al., 2005). Hong Kong, located in the Southern coastal part of China, is an important part of the Guangdong-Hong Kong-Macau Greater Bay Area (GBA), which includes the Pearl River Delta (PRD)

region in Guangdong plus Hong Kong and Macao. The Hong Kong government has been assiduous in controlling the local emission via the Air Pollution Control Ordinance, in addition to cooperating with the neighboring Guangdong and Macao governments on formulating control policies to reduce air pollution emissions in the Greater Bay Area (GBA) (HKEPD, 2021). Ambient monitoring of criteria air pollutants plays an important role in verifying the effectiveness of control policies. For



example, from 2006 to 2018, large reductions have been documented for sulfur dioxide (-81%), nitrogen dioxide (-28%), and
$PM_{10}$ (-36%) in term of annual average concentrations (HKEPD, 2019; 2020), reflecting the benefits from a series of $SO_2$, $NO_2$
and PM reduction measures (Table S8).

PM$_{2.5}$ was introduced as a criteria pollutant in Hong Kong in 2004 while its online monitoring preceded 5 years earlier at
three sites (Tap Mun, Tung Chung, Tsuen Wan) in 1999. PM$_{2.5}$ mass was added as a monitoring parameter in 2015 to the PRD
Regional Air Quality Monitoring Network, which has included 23 sites in the GBA since 2015. The monitoring data indicates
that while substantial progress has been made in lowering the pollution level of PM$_{2.5}$, from 38 µg /m$^3$ in 1999 to 15 µg /m$^3$ in
2020 (Figure 1a), the current level still notably exceeds the most updated the Air Quality Guideline of an annual average of 5
µg/m$^3$ as recommended by the World Health Organization (World Health Organization, 2021). This reality highlights the need
for continued efforts to further identify specific emission sources such that effective management strategies can be formulated.
Different from criteria gaseous pollutants, PM$_{2.5}$ is a complex mixture containing inorganic components (e.g., sulfate, nitrate,
and ammonium), elemental carbon (EC), organic carbon (OC) (consisting of tens of thousands of individual organic
compounds), and metal oxides. The accumulation of PM$_{2.5}$ pollution could come from direct emissions from human activities
and biogenic sources and/or atmospheric formation processes. Additionally, changes in air quality could be masked by
variations in atmospheric dispersion conditions on daily, seasonal, and annual bases. The multiple layers of complexity mean
that PM$_{2.5}$ mass concentration alone is insufficient to identify contributing sources or to attribute a reduction in PM$_{2.5}$ to a
particular control measure. This is evident from Figure 1b, which shows the percentage changes vary significantly among
PM$_{2.5}$ components, using the data set to be discussed in this work as an illustration.

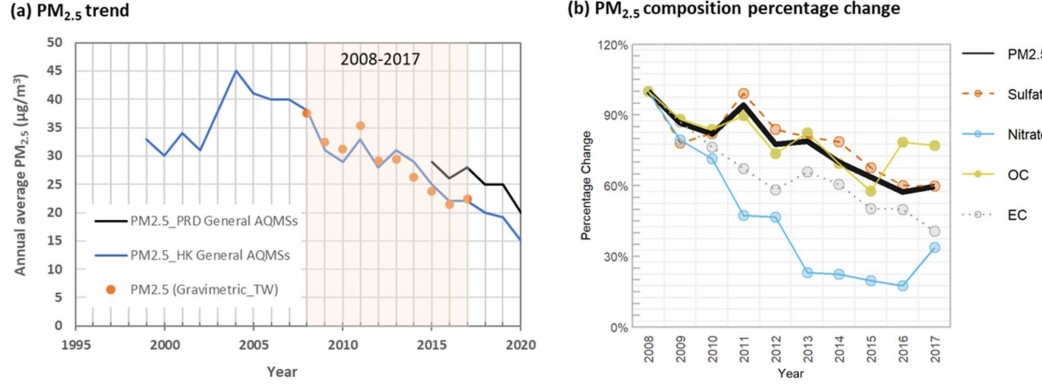

**Figure 1**. (a) Trend of annual average PM$_{2.5}$ in Hong Kong for the period of 1999-2020 and the PRD regionwide average PM$_{2.5}$ during 2015-
2020. The regionwide annual PM$_{2.5}$ is the average of 23 monitoring stations in the PRD Regional Air Quality Monitoring Network, including
three stations located in Hong Kong. The shaded period (2008-2017) has available PM$_{2.5}$ major components and select source tracers at
Tsuen Wan (TW), an urban site in Hong Kong, with the yellow dots representing gravimetrically determined PM$_{2.5}$ concentrations from the
collected filter samples at TW. (b)The percentage changes of PM$_{2.5}$ and its major composition at TW using 2008 as the base year.

The speciated analysis of PM$_{2.5}$, in particular the measurement of source-specific marker species, provides valuable
information for understanding the sources, formation, and evolution of the PM pollution. Long time-series of such chemically
specific data would, on the other hand, potentially allows the discernment of meaningful trends that are not apparent in one-
time field projects, as well as establishes long-term knowledge about representative urban/regional aerosol chemistry.
However, long-term measurements of PM$_{2.5}$ species in China are very limited, more so for the source-specific markers. Our
research team launched a filter-based PM$_{2.5}$ monitoring program in mid-2007 at Tsuen Wan (TW), an urban location, and has
maintained the operation since then. Our monitoring program adopts a regular sampling schedule of one 24-h sample every
six days to ensure temporal representativeness. High-volume samples were also collected to allow sufficient aerosol materials



for analysis of organic source markers (e.g., hopanes for vehicular emissions, and levoglucosan for biomass burning emissions). Starting from 2011, Hong Kong Environmental Protection Department (HKEPD) established a regular $PM_{2.5}$ chemical speciation monitoring network, with TW as one of its monitoring stations and adopting a 1-in-6-day sampling schedule as well (e.g., Yu et al., 2012). The field sampling, laboratory analyses, quality check/quality assurance, and data validation were conducted according to the same set of standard operating procedures which are in reference to those recommended by the US Environmental Protection Agency (Chow and Watson, 1998; USEPA, 2016). The laboratory analyses of $PM_{2.5}$ mass and major components (water-soluble ions, OC, EC, and elements) were performed by our research team except for samples from the year of 2015, which were analyzed by the Desert Research Institute (Chow et al., 2016). This set of filter samples allows us to observe the long-term trend of $PM_{2.5}$ major components and source tracers and to examine variations in aerosol sources affecting Hong Kong urban atmosphere for the 10-year period of 2008-2017.

Studies of $PM_{2.5}$ speciation data derived using a consistent sampling and analysis methodology over a period of as long as a decade and as early as 2008 are few and far between in China and elsewhere in Asia. A few multi-year studies were documented in the literature. One study covered a rural site (Wanqinsha) in the GBA in the fall and winter seasons over 6 years (2007–2012) (Fu et al., 2014). In the second study, $PM_{2.5}$ speciation covered 6 sites in Foshan, a populated city in the GBA, in winter and summer seasons over 7 years (2008–2014) (Tan et al., 2016). In the third study, $PM_{2.5}$ samples were collected in urban Beijing from 2011 to 2015 (Lang et al., 2017). Unfortunately, these long-term studies did not follow regular sampling schedules throughout the annual cycle. Some months of the year were not sampled, biasing their temporal representativeness in tracking long-term trends of $PM_{2.5}$ composition. Recently, the Chinese Central Government set up the National Aerosol Composition Monitoring Network in 2017 with a view to evaluating the effectiveness of its "2+26" strategy for improving air quality (Chen et al., 2019; Dao et al., 2019). This nationwide monitoring program is expected to generate high quality $PM_{2.5}$ composition data in the long run. Yet, a long-term data set is not available due to the limited operating period.

In this work, we analyzed the trends of $PM_{2.5}$ and its major components and individual source marker molecules/elements by the Seasonal and Trend decomposition with LOESS (STL), a robust method for extracting trend components from concentration time series (Cleveland et al., 1990), and cross-compared the results with non-parametric Mann-Kendall test and Sens's slope. The objectives are to quantify the ten-year variations in $PM_{2.5}$ chemical composition and to characterize how major local and regional sources impacting $PM_{2.5}$ pollution in Hong Kong have varied in the decade. The aim of this work is to provide a well-scrutinized long-term data set of $PM_{2.5}$ chemical composition and a sound analysis of source implications for an urban location in South China to support studies of control measure evaluation and formulation.

## 2. Data and method

### 2.1 $PM_{2.5}$ chemical speciation data

$PM_{2.5}$ filter samples were collected on a 24-hour basis (midnight to midnight) once every 6 days from 2008 to 2017 at Tsuen Wan (TW, 22o37'18N, 114o11'50E), an urban air quality monitoring station (AQMS) in Hong Kong. TW is a station surrounded by residential and commercial buildings and located about 3.3 km north to the city's international shipping port (Kwai Chung and Tsing Yi Container Terminals). Both high-volume and mid-volume samplers were equipped at the station. The high-volume sampler (Andersen Instrument, Smyrna, GA, USA) was loaded with a pre-baked 20 x 25 cm quartz fiber filter and operated at a flow of $1.13 \text{ m}^3 \text{ min}^{-1}$. Two types of mid-volume samplers were used in the course of ten years. From 2008 to 2010, a RAAS four-channel mid-volume sampler (Andersen Instrument, Smyrna, GA, USA) was operated and the





configuration of the four channels was as below: Channels 1 and 4 sampling at 16.7 L min$^{-1}$ and loaded with one 47-mm Teflon

filter and one 47-mm quartz filter, and Channels 2 and 3 sampling at a flow rate of 7.3 L min$^{-1}$ and loaded with one 47-mm nylon filter and one 47-mm quartz filter. From 2011 to 2017, two mid-volume samplers (Partisol R&P, Model 2025, Albany, NY, USA) were operated side-by-side to collect one Teflon and one quartz fiber filter at 16.7 L min$^{-1}$. A total of 592 sets of filter samples were collected, each set consisting of one 20x25cm filter and multiple 47-mm filter samples for a sampling day.

A suite of chemical speciation analysis was conducted on the collected sample filters (Table 1). Specifically, the 47-mm

Teflon filters were used to determine the PM$_{2.5}$ mass concentrations by gravimetric analysis and the trace element concentrations by energy dispersive X-ray fluorescence spectrometry (XRF). The Nylon filters from 2008-2010 and the quartz filters from 2011-2017 were used to quantify the concentrations of major ions by ion chromatography (IC). Organic carbon (OC) and elemental carbon (EC) were measured by a thermal/optical transmittance (TOT) method. Concentrations of saccharides were analyzed by high-performance anion-exchange chromatography coupled with pulsed amperometric detection

(HPAEC-PAD) (Kuang et al., 2015). Non-polar organic compounds, including alkanes, polycyclic aromatic hydrocarbons, and hopanes, were quantified by a method coupling in-injection port thermal desorption gas chromatography with mass spectrometry (TD-GC/MS) (Ho and Yu, 2004; Ho et al.,2008).

Data validation on the PM$_{2.5}$ speciation was carried out at three levels according to the publication "Guideline on Speciated Particulate Monitoring" prepared for the USEPA by Chow and Watson (1998). Level I validation mainly consists of flagging

measurements that deviate from procedures through reviewing sampling log sheets and field quality check records and identifying invalid values. Level II validation checks the internal consistency among data from different analyses, involving the following: (1) comparing a sum of measured chemical species vs. PM$_{2.5}$ mass concentrations, (2) comparing total sulfur by XRF vs. sulfate by IC, (3) comparing total potassium by XRF and soluble potassium by IC), (4) calculating anion/cation balances, and (5) examining time series data to identify and investigate outliers. Level III validation is part of the data

interpretation process, mainly focusing on identification of unusual values through parallel consistency tests with other independent datasets.

Details of analytical procedures and data validation are documented in our previous studies (Huang et al., 2014; Chow et al., 2022) and in a series of project reports (Yu et al., 2012; 2013; 2014; 2015; Yu and Zhang, 2017; 2018; Chow et al., 2016), which are available at: http://www.epd.gov.hk/epd/english/environmentinhk/air/studyrpts/pm25_study.html. See Yu et al.

(2022) for dataset access details.

**Table 1**. PM2.5 and list of components targeted for trend analysis and their measurement methods

| Species | Measurement method | Source characteristics |
|---|---|---|
| PM$_{2.5}$ | Gravimetry[1] | |
| OC | Thermal/optical analysis[2] | Primary emissions and Secondary formation with VOCs as direct precursors |
| EC | | Combustion sources |
| Sulfate | Ion chromatographic analysis of aerosol water extracts[3] | Secondary with SO$_2$ as direct precursor |
| Nitrate | | Secondary with NO$_x$ as direct precursor |
| Ammonium | | Secondary, particle presence in close association with sulfate and nitrate |
| K$^+$ | | Biomass burning, sea salt, dust |
| Al, Si | Energy dispersive X-ray fluorescence spectrometry[3,4] | Soil dust |
| Ni, V | | Residual oil combustion |
| Pb, Cu, Zn | | Coal combustion, metal industries |
| Hopanes | Thermal desorption-GC/MS[5,6] | Fossil fuel uses such as vehicular emission, residual oil burning |
| Levoglucosan | High-performance anion-exchange chromatography coupled with pulsed amperometric detection[7] | Biomass burning |

[1] USEPA, 2016; [2] Chow et al., 2007; [3] Huang et al., 2014; [4] Watson et al., 1999; [5] Ho and Yu, 2004; [6] Ho et al., 2008; [7] Kuang et al., 2015.





**2.2 Gaseous pollutant and meteorological parameter data**

Criteria gaseous pollutant data (CO, SO$_2$, O$_3$, and NO$_x$) and meteorological parameters at TW are from the network of air quality monitoring station operated by the Hong Kong Environmental Protection Department (HKEPD). The criteria gaseous pollutant data are available from the HKEPD Environmental Protection Interactive Centre (https://cd.epic.epd.gov.hk/EPICDI/air/download/) while the meteorological parameters can be retrieved from the HKUST ENVF Atmospheric & Environmental database (http://envf.ust.hk/dataview/gts/current/). The decadal time series of

temperature, relative humidity (RH), wind speed and direction, and precipitation at the site are summarized in Section S1 in supplementary document.

**2.3 Seasonal and trend decomposition with LOESS method (STL)**

For the evaluation of the overall trend of time series independently of seasonal influence, the Season and Trend decomposition (STL) was adopted to decomposes a time series ($Y_v$) into three components: the trend component ($T_v$), the

seasonal component ($S_v$) and the remainder ($R_v$) in an additive or a multiplicative manner, as per Eq. 1a and Eq. 1b. (Cleveland et al., 1990).

$$Y_v = T_v + S_v + R_v \tag{1a}$$

$$Y_v = T_v \times S_v \times R_v \tag{1b}$$

The STL algorithm is performed via locally weighted regression (LOESS) under two iterative loops. In comparison with other

time series decomposition techniques, such as the simplest Moving Averages (MA) method (Molugaram and Rao, 2017), the STL method has more flexibility in parameter tuning as well as higher robustness to counterpart the influences from outliers. A detailed description about its algorithm is provided in Section S2 in the supplementary information. The STL method has been readily implemented and widely tested in most programming languages such as *R* and Python. In this study, we utilized the STL function in the *stlplus* package in *R* for the following calculation.

Before applying the STL method, we manually inspected the data and removed data points exceeding the upper quartile by 3 times of interquartile range (i.e., $X_{75\%} + 3(X_{75\%} - X_{25\%})$) to avoid influence of extreme concentrations on the trend slope (Singh et al., 2021; Bigi and Ghermandi, 2014). The concentration data were found log-normally distributed in Q-Q plots as shown in Figures S5 & S6. Thus, log-transformation and monthly averaging were applied to create a normally distributed time series with even time interval to cope with the assumptions of the STL model.

**2.4 Generalized Least Squares with Autoregressive-Moving Average (GLS-ARMA) model**

The trend curves from STL method are often too irregular to be described verbally or quantitatively. This prompts us to seek a method that allows calculation of an overall changing rate for the trend component for further analysis. When dealing with time series data with autocorrelation (i.e., the current value ($Y_v$) depends on its lagged values ($Y_{v-h}$)), generalized least squares (GLS), instead of ordinary least squares (OLS), is more suitable for the quantification of changing rate of the time

series. In GLS, the covariance matrix (and so the residuals) can be estimated by an Autoregressive-Moving-Average ($ARMA(p,q)$) model. Specifically, the $ARMA(p,q)$ model assumes that the current value ($X_t$) is influenced by its $p$-order of lagged values ($X_{t-h}$) and $q$-order of lagged residuals ($\varepsilon_{t-i}$) as shown in Eq. 2.

$$X_t = \sum_{h=1}^{p} \phi_h X_{t-h} + \varepsilon_t + \sum_{i=1}^{q} \theta_i \varepsilon_{t-i} \tag{2}$$

The determination of $p$ and $q$ in an $ARMA(p,q)$ model is achieved by minimizing model selection criteria such as Akaike's

Information Criterion (AIC) and Bayesian Information Criterion (BIC). A model with a smaller AIC or BIC value is deemed





more likely to generate the data that we obtain, while taking both probability likelihood and model simplicity into consideration. Despite the differences in assumptions, theoretically AIC and BIC give similar results in most cases (Shumway and Stoffer, 2017). In our study, we calculated AIC, AICc (a bias corrected AIC), and BIC. As tabulated in Table S1 in in the supplementary information, these criteria indictors show no significant difference in terms of the parameter selection outcomes. Thus, hereafter only BIC values are reported when determining the slope of each GLS-ARMA model analysis. The details of the methodology of this hybrid STL-GLS-ARMA method are provided in the Section S3 in supporting document.

**2.5 Comparison with other trend analysis methods**

Additional trend analysis methods were explored to cross-validate the results from the GLS-ARMA method. First, Sen's slope method (Wilcox, 2017), a non-parametric method, was performed on the same dataset to calculate the changing rate. In Sen's slope method, we first run Mann-Kendall test to see whether the overall trend of the annual averages is monotonic. Then the median of the slopes for all pairwise data points are computed and defined as the Sen's slope. Second, an exponential trend estimation was computed using Compound Annual Growth Rate (CAGR) for each pairwise combination of annually averaged values using Eq. (3).

$$CAGR\ (\%) = \left(\frac{X_{t_n}}{X_{t_0}}\right)^{\frac{1}{t_n - t_0}} \times 100\% \tag{3}$$

where $X_{tn}$ and $X_{t0}$ is the annual averaged time series at time $t_n$ and $t_0$, respectively. Like Sen's slope method, the overall trend of a species is represented by the median value of all the CAGR results. Testing the agreement between the linear and non-linear approaches helps in validating the overall trend analysis results. Details of CAGR results are provided in the Section S7 of supporting information.

**3. Results and discussion**

**3.1 PM$_{2.5}$ composition**

PM$_{2.5}$ and its major components collected over the decade are displayed in time series of monthly averages in Figure 2a and annual averages in Figure 2b. The time series of individual samples are provided in Figure S3. Under influence of the monsoon winds, the four seasons in Hong Kong are well distinguished in their meteorological characteristics, with summer and winter being the two longest seasons and each lasting approximately four months. The four seasons are approximately spring from 16 Mar to 14 May, summer from 15 May to 15 Sep, fall from 16 Sep to 15 Nov, and winter from 16 Nov to 15 Mar (Chin, 1986). Under the influence of the Asian monsoon, the northly prevailing winds carry dry and polluted northern continental air masses to Hong Kong in wintertime whereas prevailing southerly and southeasterly monsoon winds in summertime and bring largely clean marine air masses from South China Sea or Northwest Pacific Ocean. As a result, PM$_{2.5}$ and other pollutants show distinct winter-summer contrast in their source origins and in concentration levels (Yu et al., 2004). In summer, local emissions have dominant influence while in winter, the regional/superregional pollution significantly elevated air pollutant levels. We thus show separate time series for summer and winter seasonal average PM$_{2.5}$ chemical composition in Figures 2c and 2d and discussion of the source trend according to seasons of summer and winter provides a more direct understanding of source variations over the years. Spring and fall, being two short and transient seasons, display more variable and mixed influences from both local and regional/superregional sources (Figure S4). Their time series are less useful for tracking decadal source variations, therefore not discussed in this paper.

As shown in Figure 2, an overall decline trend is clearly seen in both bulk PM$_{2.5}$ and its major components over the decade

of 2008-2017. Sulfate and organic matter (OM) remain to be the top two dominant PM$_{2.5}$ components throughout the decade and for both winter and summer seasons. Significant monthly variations are also evident, with highest concentrations in the winter months and the lowest in the summer months. The highest winter month average could be more than double the lowest

215    summer average concentration in a same year, clearly indicating the significant contribution of regional/superregional pollution to PM$_{2.5}$ in Hong Kong. Comparing Figures 2c and 2d, we see that the mass reductions in summer season over the decade are much less in comparison with those seen for the winter, however, a continuous decline in EC is clear in the decade-long time series of the summer averages, indicating success in controlling local EC sources (i.e., vehicular emissions). A quantitative description of 10-year trends for PM$_{2.5}$, its major components and source tracers will be provided in the ensuing sections.

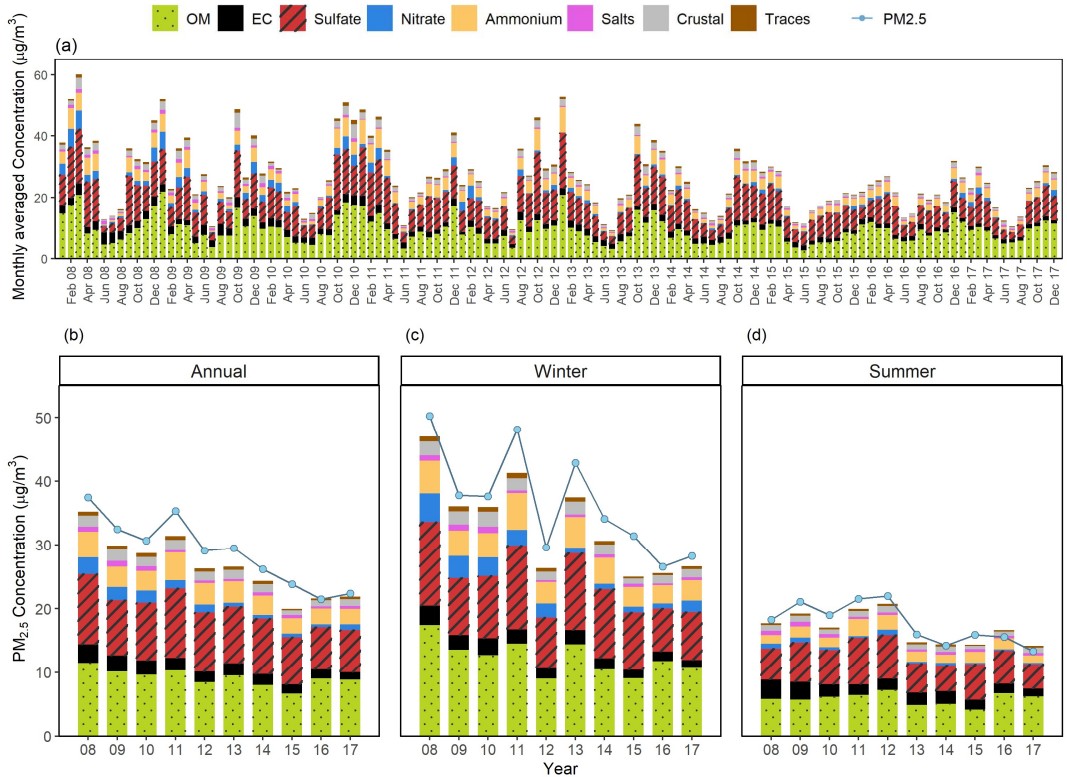

**Figure 2.** Time series of PM$_{2.5}$ chemical composition from 2008 to 2017 in the form of (a) monthly averages, (b) annual average, (c) winter seasonal averages, and (d) summer seasonal averages. In the legends, "Salts" includes Na$^+$ and Cl$^-$; "Crustal" represents crustal materials, computed to be 1.89*Al+2.14*Si+1.4*Ca+1.43*Fe; and "Tracers" includes elements other than Na, Cl, S, K, Al, Si, Ca, Fe. OM refers to organic matter and computed to be 1.4*OC.

225    For a simple illustration of the 10-year change in PM$_{2.5}$ chemical composition, the average chemical compositions in the starting and the ending year of the decade are compared in Figure 3. On the annual average basis, the top four major components remain to be the same, i.e., OM, sulfate, ammonium, and nitrate, collectively accounting for a comparably ~84% of PM$_{2.5}$ in 2008 and 2017, despite 10 years apart. Among the four top contributors, OM has gained a few percent while nitrate has been reduced by a few percent in proportional importance. The 10-year compositional changes are more prominent in the seasonal

230    averages. For winter PM$_{2.5}$, the relative importance of OM increased (up from 31% in 2008 to 40% in 2017) while the relative abundance of nitrate decreased (down from 7% to 4%), as well as EC (down from 7.8% to 5.3%). For summer PM$_{2.5}$, the most





significant compositional changes are also OM (up from 32% to 44%), EC (down from 16.7% to 8.5%), and nitrate (down from 4.3% to 1.7%). The proportional decrease of EC was most notable, reflecting the effectiveness of local vehicular emissions control measures.

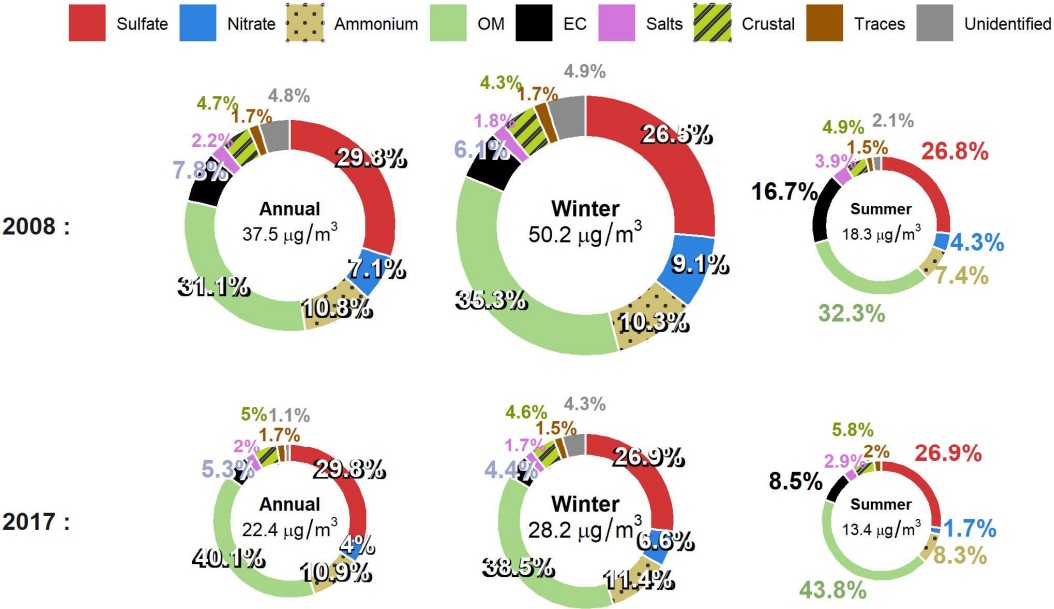

**Figure 3**. Comparison of average $PM_{2.5}$ compositions between the starting year (2008) and the ending year (2017) of the decade. In each year, three averages are shown, corresponding to annual, winter, and summer, and the donut size is proportional to the $PM_{2.5}$ concentration.

### 3.2 Annual trend analysis

Previous studies that examined annual trend of pollutants for evaluation of pollutants reduction in Hong Kong adopted a simple method of comparing annual average values (HKEPD, 2020; Zhang et al., 2018; Lu et al., 2013; Yuan et al., 2013). While this approach avoids the autocorrelation issue ─ the lag value of variables ($Y_{t-h}$) influences the current value ($Y_t$) in time series─ it would suffer increased bias due to the sacrifice of the sample size for estimation. In comparison, STL is a more robust method for extracting trend components from concentration time series (Cleveland et al., 1990), with the autocorrelation issue accounted for by GLS-ARMA (Shumway and Stoffer, 2017). The STL-GLS-ARMA method has been adopted in a few studies analyzing air pollutant trends (e.g., Anttila and Tuovinen, 2010; Bigi and Ghermandi, 2014). It is found that that STL-GLS-ARMA has the advantage of retaining more degree of freedom on sample population and thus producing a more accurate estimate than the ordinary least squares (OLS) method.

We applied STL-GLS-ARMA to the monthly average concentrations of $PM_{2.5}$ mass and individual species, including major components ($SO_4^{2-}$, $NO_3^-$, $NH_4^+$, OC, and EC), and source-specific molecular or elemental tracers (i.e., $K^+$, Al, Si, V, Ni, Pb, Zn, Cu, hopanes, and levoglucosan), as well as the routinely monitored criteria gaseous pollutants (CO, $SO_2$, $NO_x$ and $O_3$) (Figure 4). Table 2 summarizes the slopes obtained from the GLS-ARMA, Sen's slope method, and percentage change of each species over 2008-2017, together with the annual average concentration data in 2008 and 2017. The results from both the slope-determining methods were in good agreement for all the $PM_{2.5}$ species. The GLS-ARMA trend slopes are significantly less than zero at a *p* level of <0.001 for all $PM_{2.5}$ measurement parameters, except for V and levoglucosan, which are significant





255 at a higher $p$ level (0.01 and 0.05, respectively). For the Sen's slopes, they are less than zero at a lower $p$ level of 0.01 for most species and at $p = 0.05$ for levoglucosan, and not significant at $p > 0.05$ for V and hopanes. Such differences reflect the superiority of the GLS-ARMA method arising from retaining more degrees of freedom on the sample population. Thus, we'll adopt the GLS-ARMA slopes in commenting the ten-year changing rate in later discussion.

 From the results of STL-GLS-ARMA method, a declining rate of 1.5 µg m$^{-3}$ per year was estimated for the PM$_{2.5}$ mass.

260 This decline was significantly attributed by the top two major components, namely sulfate accounting for 24% (Slope: -0.36 µg m$^{-3}$ yr$^{-1}$) and OM 17% (-0.18*1.4 = -0.25 µg m$^{-3}$ yr$^{-1}$), respectively. Ammonium, nitrate, and EC decreased in a similar rate in mass concentration unit (NH$_4^+$, NO$_3^-$, EC: -0.12, -0.17, -0.17 µg m$^{-3}$ yr$^{-1}$), they accounted for a similar percentage at around 8.0-11% each and a combined 31% of the overall PM$_{2.5}$ reduction. Meanwhile, other components such as biomass burning markers (K$^+$ and levoglucosan), industrial and coal combustion tracers (Zn and Pb), crustal materials (Al, Si, and Ca),

265 altogether explain the remaining 28% of PM$_{2.5}$ depletion. Note that the tracer species only account for a minute amount of mass, however, they are indicative of other unmeasured PM$_{2.5}$ components co-emitted with these sources.

 The percentage changes during the decade are calculated using the annual average in 2008 and 2017 and listed in Table 2. With the GLS-ARMA model fitted data, we can also calculate the percentage changes. Comparing the two approaches, the GLS-ARMA method yields higher percentage drops in K$^+$, NO$_3^-$, Al, Si, Pb, and Cu than those calculated using annually

270 averaged data. This could be explained by the different concentration levels fitted by GLS-ARMA model. The underestimated concentration in 2017 by GLS-ARMA results from the flatten variation in the later years (Figure 4), hence the higher percentage changes in these species. This problem was less obvious for species with smoother declines such as SO$_2$, NO$_x$, OC, EC, V, Ni, and hopanes. Therefore, the differences in percentage change between two methods helped on identifying the different changing characteristics along the time series.

275 The annual percent change rates computed using CAGR, summarized in Figure S11 and Table S5, show a good agreement with those from the linear approaches (i.e., GLS-ARMA and Sen's slope). In general, the exponential approach of the CAGR method estimates a larger decline than the GLS-ARMA method. The maximum difference occurs with NO$_3^-$ (-10%). The relative constant concentration levels in the later years were particularly observed in NO$_3^-$, which would imply a faster reduction in an exponential variation model and thus result in a larger discrepancy. The absolute differences for all the other

280 species are less than 5%. For simplicity, we confine trend discussion to results from the linear approaches.

 The 10-year percentage change in PM$_{2.5}$ is -40%. Sulfate and ammonium, with nearly identical decrease trajectories due to their close chemical linkage, have their percentage drops closely matching that of bulk PM$_{2.5}$ (Figure 1b and Table 2). Other major components, however, differ in their percentage reductions from that of bulk PM$_{2.5}$, with the reduction in nitrate (-66%) and EC (-60%) exceeding while OC (-23%) falling below that of bulk PM$_{2.5}$. Such results reveal the effectiveness of control

285 measures in lowering EC and sulfate and the increasing importance of OC in addressing PM$_{2.5}$ pollution in the coming years.

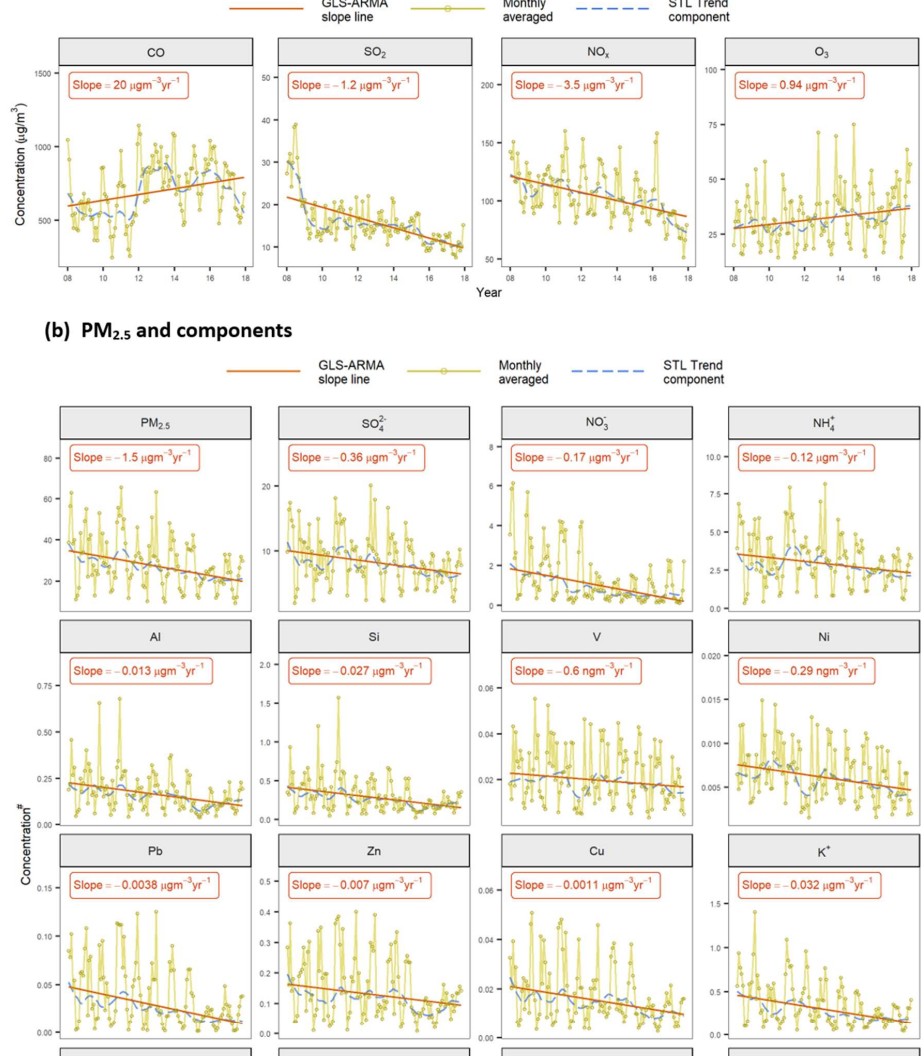

**Figure 4.** (a) Gaseous and (b) PM$_{2.5}$ pollutant data over 2008-2017: monthly concentrations (blue), trend component (cyan), and the slope line of trend determined by GLS-ARMA method (red). Note that the concentrations of hopanes and levoglucosan are in ng m$^{-3}$, while the others are in µg m$^{-3}$.


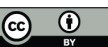



**Table 2.** Summary of the decadal trend slopes and percent changes from 2008 to 2017 obtained with the GLS-ARMA analysis of the de-seasonalized time series and with the Sen's slope from annual averaged time series.

| Species | unit | Concentration | | Slope | | | %Relative change (2008-2017) | | |
|---|---|---|---|---|---|---|---|---|---|
| | | 2008 | 2017 | ARMA model[1] | GLS-ARMA[2] | Sen's slope[2] | GLS-ARMA | Annual-averaged | Difference[3] |
| **Gaseous pollutants** | | | | | | | | | |
| CO | µg m$^{-3}$ | 621 | 662 | ARMA (1,0) | 20, (-1.5, 41) | 18, (-16, 48) | 28% | 6.6% | 21% |
| SO$_2$ | µg m$^{-3}$ | 28.4 | 10.7 | ARMA (1,0) | -1.2***, (-1.8, -0.65) | -0.71**, (-1.6, -0.29) | -51% | -62% | 11% |
| NO$_x$ | µg m$^{-3}$ | 121 | 77 | ARMA (1,0) | -3.5***, (-4.5, -2.4) | -2.9*, (-4.9, -1.5) | -26% | -36% | 10% |
| O$_3$ | µg m$^{-3}$ | 30.9 | 41.6 | ARMA (1,0) | 0.94***, (0.41, 1.5) | 0.82*, (0.26, 1.4) | 30% | 35% | -5% |
| **Particle pollutants** | | | | | | | | | |
| PM$_{2.5}$ | µg m$^{-3}$ | 37.5 | 22.4 | ARMA (1,0) | -1.5***, (-1.9, -1.1) | -1.6**, (-2.3, -1.3) | -40% | -40% | 0% |
| SO$_4^{2-}$ | µg m$^{-3}$ | 11.0 | 6.60 | ARMA (0,2) | -0.36**, (-0.58, -0.15) | -0.43*, (-0.72, -0.19) | -33% | -40% | 7% |
| NO$_3^-$ | µg m$^{-3}$ | 2.64 | 0.91 | ARMA (1,0) | -0.17***, (-0.21, -0.12) | -0.23**, (-0.33, -0.073) | -85% | -66% | -19% |
| NH$_4^+$ | µg m$^{-3}$ | 3.99 | 2.41 | ARMA (2,0) | -0.12*, (-0.22, -0.027) | -0.16**, (-0.31, -0.038) | -32% | -40% | 8% |
| OC | µgC m$^{-3}$ | 8.22 | 6.33 | ARMA (2,0) | -0.18*, (-0.31, -0.042) | -0.22*, (-0.43, -0.082) | -23% | -23% | 0% |
| EC | µgC m$^{-3}$ | 2.95 | 1.18 | ARMA (0,2) | -0.17***, (-0.2, -0.13) | -0.16***, (-0.26, -0.12) | -56% | -60% | 4% |
| Al | ng m$^{-3}$ | 223 | 140 | ARMA (1,0) | -13***, (-17, -8) | -12**, (-21, -8.5) | -51% | -37% | -14% |
| Si | ng m$^{-3}$ | 412 | 222 | ARMA (1,0) | -27***, (-35, -18) | -30**, (-40, -19) | -60% | -46% | -14% |
| V | ng m$^{-3}$ | 23.6 | 15.7 | ARMA (1,0) | -0.60*, (-1.1, -0.14) | -0.53, (-1.4, 0.14) | -24% | -34% | 10% |
| Ni | ng m$^{-3}$ | 6.94 | 4.46 | ARMA (1,0) | -0.29***, (-0.42, -0.16) | -0.29*, (-0.47, -0.11) | -35% | -36% | 1% |
| Pb | ng m$^{-3}$ | 56.6 | 18.8 | ARMA (1,0) | -3.8***, (-5, -2.7) | -4.7**, (-6.7, -3.5) | -75% | -67% | -8% |
| Zn | ng m$^{-3}$ | 185 | 112 | ARMA (2,2) | -7.0**, (-12, -2) | -9.5*, (-16, -2.9) | -39% | -40% | 1% |
| Cu | ng m$^{-3}$ | 20.6 | 11.7 | ARMA (1,0) | -1.1***, (-1.7, -0.6) | -1.4**, (-2.1, -0.65) | -50% | -43% | -7% |
| K$^+$ | ng m$^{-3}$ | 556 | 223 | ARMA (1,0) | -32***, (-41, -23) | -40**, (-52, -24) | -65% | -60% | -5% |
| Hopanes | ng m$^{-3}$ | 0.708 | 0.180 | ARMA (1,0) | -0.052***, (-0.075, -0.03) | -0.041, (-0.06, 0.015) | -71% | -75% | 4% |
| Levoglucosan | ng m$^{-3}$ | 61.3 | 31.7 | ARMA (0,1) | -1.4*, (-2.8, -0.089) | -2.6*, (-4.9, -1.2) | -31% | -47% | 16% |

[1] The optimal ARMA model parameters were selected by BIC. [2] Numbers in the bracket denote the 95% confident interval for the slope value while asterisks denote that the slope significantly differs from zero: * $p < 0.05$, ** $p < 0.01$, *** $p < 0.001$. [3] The difference between the two methods (GLS-ARMA – Annual-averaged).




### 3.3 Trend analysis of winter and summer data

As discussed in Section 3.1, $PM_{2.5}$ levels and dominant sources are distinctly different in winter and summer. The two seasons merit separate analysis of their ten-year trends. This is further supported by correlation and hierarchical clustering analysis of year-by-year data, the results of which are shown in Figure 5 using 2008 and 2017 as examples. Figure 5 reveals

that the measurement variables segregate into two clusters marked in black and pink linkage lines, respectively, and they broadly correspond to one group of pollutants known to be significantly influenced by regional/super-regional sources (e.g., OC, sulfate, nitrate, $NH_4^+$, $K^+$, Pb, Zn, Cu) and a second group of species with dominant contributions from local sources (i.e., NOx, EC, hopanes, Ni, and V). The regional sources have strong seasonality under the influence of the monsoon winds.

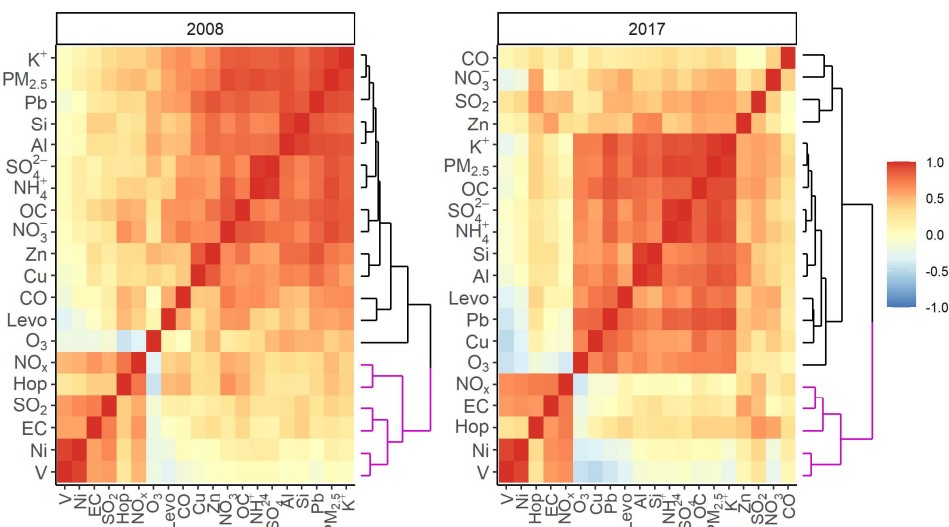

**Figure 5.** Correlation matrix of gaseous and particulate pollutants with hierarchical clustering results at 2008 (left) and at 2017 (right). Clusters: pink – local sources; black – regional sources.

Figure 6 shows the 10-year variations of the seasonal average concentrations of $PM_{2.5}$ and select components for winter and summer. The season-specific Sen's slopes are listed in Table 3, expressed in both mass concentration change rate per year and percent change rate per year. The latter unit allows a direct comparison of relative source strength changes of local and

regional sources by removing the impact of meteorological factors (e.g., boundary layer height) on ambient concentrations. Seen in Table 3, the Sen's slope for bulk $PM_{2.5}$ is significantly different seasonally, at -2.0 μg m$^{-3}$ yr$^{-1}$ in winter vs. -0.67 μg m$^{-3}$ yr$^{-1}$ in summer, while the percentage decline rates are comparable, at -3.9% yr$^{-1}$ in winter and -3.7% yr$^{-1}$ in summer. In align with the species segregation revealed in Figure 5, the group of regional species shows significantly larger decrease rates in mass concentration in winter than in summer, but the group of local species (EC, V, and Ni) displays comparable Sen's

slope in both seasons. It is worth noting that summer OC does not show a discernable increase or decrease trend over the decade, but winter OC shows a decrease trend with a slope of -0.45 μgC m$^{-3}$ yr$^{-1}$ (Figure 6). Such a stark contrast indicates a significant seasonal difference in OC sources and their underlying driving factors. This also implies that measures to lower the OC contribution in $PM_{2.5}$ must consider the strong seasonality of its sources.

Considering the diverged seasonality among major components and source tracers, we individually examine in the

subsequent sections the trend characteristics of the major $PM_{2.5}$ constituents and important sources that have effective tracer data.





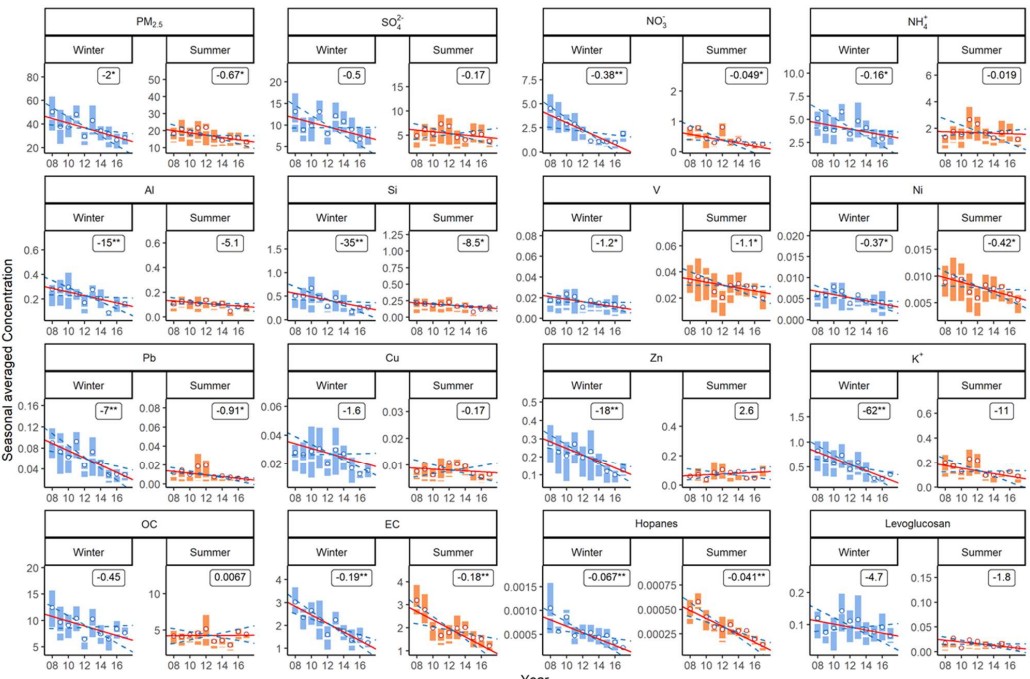

**Figure 6.** Ten-year variations of seasonal average concentrations of $PM_{2.5}$ and select components for the winter (blue) and summer (orange) from 2008 to 2017. Red lines indicate the Sen's slope, blue dashed lines indicate the 95% confidence intervals.


**Table 3 –** Summary of the seasonal variation estimated by Sen's slope.

| Species | Sen's slope (mass concentration)[1] | | | Sen's slope (percent change)[2] | |
| --- | --- | --- | --- | --- | --- |
| | Unit | Winter | Summer | Winter | Summer |
| **Gaseous pollutants** | | | | | |
| CO | $\mu g\ m^{-3}\ yr^{-1}$ | +8.9 | +21 | +1.1% | +4.4% |
| $SO_2$ | $\mu g\ m^{-3}\ yr^{-1}$ | -1.0* | -0.98*** | -3.9%* | -2.8%*** |
| $NO_x$ | $\mu g\ m^{-3}\ yr^{-1}$ | -4.4** | -3.7** | -3.3%** | -3.1%** |
| $O_3$ | $\mu g\ m^{-3}\ yr^{-1}$ | +1.6* | +0.56 | +4.9%* | +2.6% |
| **$PM_{2.5}$ and its components** | | | | | |
| $PM_{2.5}$ | $\mu g\ m^{-3}\ yr^{-1}$ | -2.0* | -0.67* | -3.9%* | -3.7%* |
| $SO_4^{2-}$ | $\mu g\ m^{-3}\ yr^{-1}$ | -0.50 | -0.17 | -3.8% | -3.5% |
| $NO_3^-$ | $\mu g\ m^{-3}\ yr^{-1}$ | -0.38** | -0.049* | -8.4%** | -6.3%* |
| $NH_4^+$ | $\mu g\ m^{-3}\ yr^{-1}$ | -0.16* | -0.019 | -3.2%* | -1.4% |
| OC[3] | $\mu gC\ m^{-3}\ yr^{-1}$ | -0.45 | 0.0067 | -3.6% | +0.16% |
| EC | $\mu gC\ m^{-3}\ yr^{-1}$ | -0.19** | -0.18** | -6.2%** | -5.8%** |
| Al | $ng\ m^{-3}\ yr^{-1}$ | -15** | -5.1 | -5.3%** | -4.8% |
| Si | $ng\ m^{-3}\ yr^{-1}$ | -35** | -8.5* | -7.0%** | -4.1%* |
| V | $ng\ m^{-3}\ yr^{-1}$ | -1.2* | -1.1* | -7.0%* | -3.4%* |
| Ni | $ng\ m^{-3}\ yr^{-1}$ | -0.37* | -0.42* | -5.9%* | -4.7%* |
| Pb | $ng\ m^{-3}\ yr^{-1}$ | -7.0** | -0.91* | -7.7%** | -7.8%* |
| Zn | $ng\ m^{-3}\ yr^{-1}$ | -18** | +2.6 | -6.6%** | +4.4% |
| Cu | $ng\ m^{-3}\ yr^{-1}$ | -1.6 | -0.17 | -5.8% | -2.3% |
| $K^+$ | $ng\ m^{-3}\ yr^{-1}$ | -62** | -11 | -7.5%** | -5.5% |
| Hopanes | $ng\ m^{-3}\ yr^{-1}$ | -0.067** | -0.041** | -6.4%** | -8.1%** |
| Levoglucosan | $ng\ m^{-3}\ yr^{-1}$ | -4.7 | -1.8 | -3.9% | -10% |

[1]Asterisks in the table denote that the slope significantly differs from zero: * $p < 0.05$, ** $p < 0.01$, *** $p < 0.001$.
[2]The Sen's slopes in these two columns are obtained on normalized concentrations against those in 2008, thus providing percentage change rates relative to 2008, with the unit of % $yr^{-1}$.
[3]The Sen's slope for wintertime OC is significant at a $p$ level of 0.11.





### 3. 4 Secondary inorganic aerosol components

The three secondary inorganic aerosol components, namely, sulfate, nitrate, and ammonium, are constantly prominent components of $PM_{2.5}$ and make up 43-47% of $PM_{2.5}$ mass over the decade. Their ambient abundances exhibit a strong seasonality, with the winter concentrations more than double the summer concentrations. Seasonally, the wintertime levels

changed by -0.50, -0.38, and -0.16 µg m$^{-3}$ yr$^{-1}$ in mass concentration change rate and at -3.8%, -8.4%, and -3.2% yr$^{-1}$ in percentage change rate for sulfate, nitrate, and ammonium, respectively. The summertime level changed by -0.17, -0.05, and -0.02 µg m$^{-3}$ yr$^{-1}$ in mass concentration change rate and at -3.5%, -6.3%, and -1.4% yr$^{-1}$ in percentage change rate for sulfate, nitrate, and ammonium, respectively (Table 3). They are significant drivers of $PM_{2.5}$ decline.

While the direct precursor for sulfate is $SO_2$, the reduction of $SO_2$ does not necessarily translates to proportional reduction

in sulfate, as various oxidants (e.g., hydroxyl radical, hydrogen peroxide, ozone, etc.) participate in the oxidation formation of sulfate from $SO_2$ and the role of each oxidant is highly dynamic in both temporal and spatial scale (e.g., Xue et al., 2019). Nevertheless, it is informative to compare the changing rates of $SO_2$ and sulfate. Over the decade, sulfate dropped by 40% in annual average concentration, lagging behind the 62% drop recorded for $SO_2$ (Table 2). A close examination of the 10-year time series of monthly concentrations of $SO_2$ and sulfate side by side (Figure 7) shows temporally uneven reduction. The

steepest drop in $SO_2$ occurred in 2008-2009 (from 28.4 to 15.6 µg/m$^3$, a reduction of 45%), following the mandated switch to ultra-low S (<0.005%by weight) for all commercial and industrial processes in 2008. During the same period, sulfate dropped by 22% from 11.0 µg/m$^3$ in 2008 to 8.6 µg/m$^3$ in 2009. Between 2009-2014, both $SO_2$ and sulfate dropped by a same small percent (~4%) and varied in a narrow range of 14.6-16.2 µg/m$^3$ for $SO_2$ and 8.66-9.03 µg/m$^3$ for sulfate. Between 2015-2017, the introduction of $SO_2$ reduction measures targeting power plants and shipping industry led to a decrease of $SO_2$ by 22%

(from 13.8 to 10.7 µg/m$^3$) while sulfate only dropped by 11% (from 7.45 to 6.60 µg/m$^3$) (Figure 7). Evidently, the discrepant changing rates of ambient $SO_2$ and sulfate confirms that sulfate reduction is generally not proportional to local $SO_2$ reduction because of nonlinear formation chemistry of sulfate and a significant contribution to sulfate from regional transport (Chen et al., 2021; Chow et al., 2022).

The very origin of $PM_{2.5}$ $NH_4^+$, i.e., reaction of ammonia with sulfate aerosol, dictates its close association with sulfate.

This relationship is expectedly confirmed in the excellent correlation of $NH_4^+$ with sulfate in all the years (Figure 5). As $NH_3$ is generally abundantly supplied, the variation of $NH_4^+$ closely tracks that of sulfate, as confirmed in our dataset.

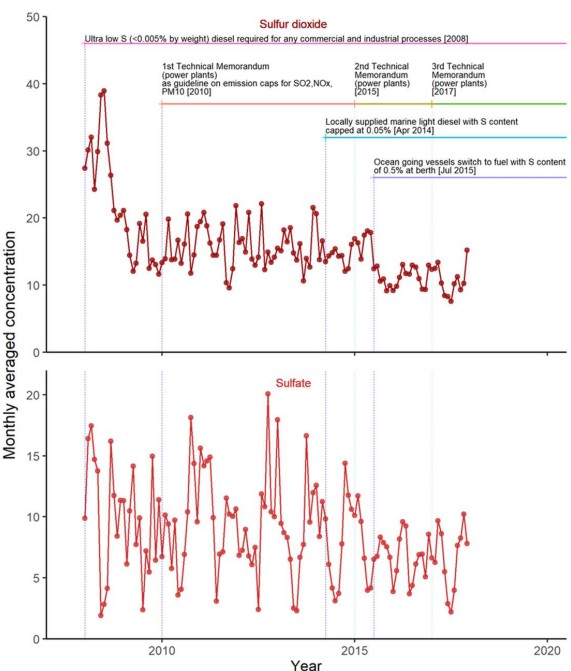

**Figure 7.** Ten-year variation of monthly SO$_2$ and sulfate concentration. The significant SO$_2$ emission control measures implemented in Hong Kong are indicated in the top plot. The unit of concentration is µg m$^{-3}$.

It is well established that PM$_{2.5}$ nitrate is a secondarily formed product from NO$_x$ oxidation (e.g., Griffith et al., 2015). Like the formation of sulfate, the involvement of multiple oxidants (e.g., hydroxy radical, O$_3$) creates significant complexity so that a proportional relationship is not expected between variations of NO$_x$ and nitrate (Xue et al., 2014a). Additionally, atmospheric physical conditions, such as temperature and RH, also strongly influence the partitioning of nitrate between gas and particle phase. Comparing the reduction rates of NOx and nitrate, we note that over the decade nitrate dropped by 66%,

higher than the reduction rate of 36% for NO$_x$ (Table 2). While the deviation from proportionality reflects the nonlinear formation chemistry of nitrate, the higher reduction rate in nitrate is seemingly counter-intuitive. Unlike sulfate, which predominantly exist in the particle-phase, nitrate could be either present as nitric acid in the gas phase or as ammonium nitrate partitioning between gas-particle phases. Additional, nitrate could significantly partition to the coarse particles (PM$_{2.5-10}$) (Xue et al., 2014b). Thus, PM$_{2.5}$ nitrate, mainly existing in the form of ammonium nitrate, only represents a fraction of the total

nitrate. This provides possibility for higher PM$_{2.5}$ nitrate reduction rate than its precursor NO$_x$. Zhang et al (2018) examined the PM$_{10}$ chemical speciation data in Hong Kong that spans 18 years (1998-2015) and found nitrate in PM$_{10}$ increased from 2002 to 2011 then decreased afterwards. Such an observation indirectly indicates that the significant presence of nitrate in coarse PM could lead to divergent trends of nitrate in PM$_{10}$ and PM$_{2.5}$. A more detailed consideration with the aid of modeling would be needed in order to reveal the variation extent of total nitrate and the distribution of different nitrate forms. Such an

exploration requires efforts going beyond the current project, thus not pursued.

**3.5 Components dominated by Local emissions – Vehicular and Shipping emissions**

    It has been recognized that on-road vehicles and marine vessels are two major local emission sources for ambient PM$_{2.5}$ in Hong Kong (Guo et al., 2009; Li et al., 2012; Cheng et al., 2015; Chow et al., 2022). A steadily decreasing trend was



observed in the concentration levels of typical vehicular emission tracers: EC and hopanes. Over the ten-year period, annual
average EC and hopanes decreased by 60% and 75% in mass concentration and at a rate of -0.17 µgC m$^{-3}$ yr$^{-1}$ and -0.052 ng
m$^{-3}$ yr$^{-1}$, respectively. These significant reductions indicate the effectiveness of an array of control measures that have been
implemented by the government since 2008 (Figure 8). Most notably, they include: (1) replacing pre-Euro IV diesel
commercial vehicles with higher Euro standards vehicles since 2007, (2) implementing the Statutory Ban against idling of
motor vehicle engines in 2011, and (3) the imposition of the emission control for petrol and LPG vehicles in 2014.  It is worth
noting that the vehicular traffic local to the sampling site has increased by~20% over the decade if we use the traffic flow
count through Shing Mun Tunnel, a tunnel less than 5 km away from the site, as an indicator (Figure S3). Despite the increase
of vehicles on the road, the decrease of ambient EC and hopanes is unambiguous, which serves as strong evidence for the
effectiveness of vehicular emission controls. On a separate yet relevant note, Wang et al. (2018) sampled and compared both
gaseous and particulate pollutants from fresh vehicular emissions in Shing Mun Tunnel in 2003 and 2015 and found that OM
and EC decreased by -70% and -80% from 2003 to 2015, respectively. This adds another measurement-based evidence for the
overall decrease in PM$_{2.5}$ burden from local vehicular emissions.

The concentration levels of shipping emission tracers (V and Ni) were reduced by 34% and 36% by mass concentration
and at a rate of -0.60 and -0.29 ng m$^{-3}$ yr$^{-1}$, respectively. The percent reduction of V and Ni is less than those of vehicular
emission tracers because their decreasing trends were not obvious until 2015 (Section S5) when shipping emission control
policy was first introduced in Hong Kong to reduce SO$_2$ emission (HKEPD, 2021).

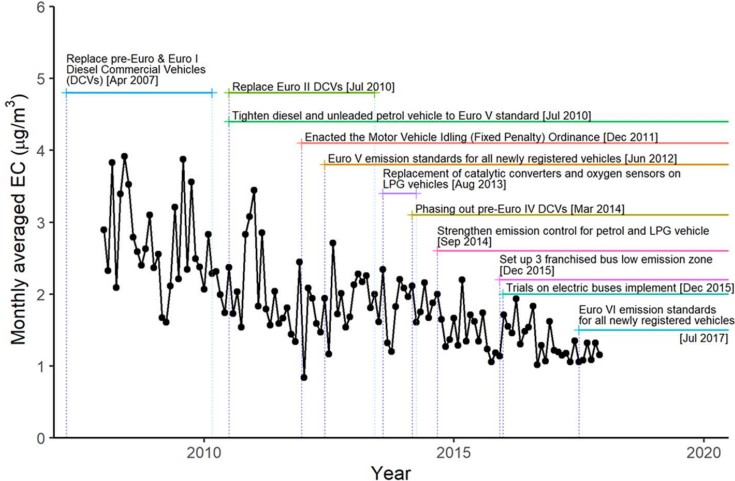

**Figure 8.** Ten-year variation of monthly EC concentration. The significant vehicular emission control measures implemented in Hong Kong
are indicated in the top plot.

### 3.6 Species significantly influenced by regional emission sources –Biomass burning, industrial/coal combustion,
**and dust sources**

Biomass burning, industrial/coal combustion, and dust are well-recognized regional sources that influence PM pollution
in Hong Kong (e.g., Zhang et al., 2018; Chow et al., 2022). Our source apportionment study of PM$_{2.5}$ at six sites in Hong Kong
in 2015 show that the combined industrial and coal combustion accounted for 12-20%, biomass burning 2-13%, and dust 4-
8% of PM$_{2.5}$ (Chow et al., 2022). The marker chemicals for these sources are among the chemical composition data monitored,
allowing us to track the long-term trend of these sources.

Levoglucosan, an abundant primary product formed during pyrolysis of cellulose, is a highly specific tracer of biomass



burning emissions (Simoneit et al., 1999). $K^+$ is also abundantly emitted from biomass burning, especially crop residue burning. In studies without levoglucosan data, $K^+$ is frequently used as a biomass burning tracer. However, $K^+$ is a less specific tracer, due to contributions from other sources such as coal combustion, dust, and sea salt (e.g., Yu et al., 2018; Chow et al., 2022).

The clustering analysis results show $K^+$ and levoglucosan were moderately correlated and fell into two different clusters in the same group of regional origin (Figure 5). Comparing the 10-year variations of these two tracers, we found that their reduction extents differed significantly, with $K^+$ at -60% and levoglucosan at -47% over the decade. When examined seasonally (Figure 6 and Table 3), they showed more distinct differences. Specifically, wintertime $K^+$ showed a definitive decline trend at a rate of -7.5% $yr^{-1}$ (p<0.01) while the decline of wintertime levoglucosan (-3.9% $yr^{-1}$) could not be discerned from zero according

to the statistical test at p<0.05 (Table 3). The lack of a clear declining trend of wintertime levoglucosan could be visually verified in Figure 6. Further, both summertime $K^+$ and levoglucosan did not show a clear decreasing trend either (Figure 6 or Table 3). The inconsistency between $K^+$ and levoglucosan could be explained if one considers that they track different types of biomass burning (i.e., $K^+$ more representative of burning crop residues high in $K^+$ content vs. levoglucosan representative of burning of cellulose, thus all types of vegetative biomass including hill fires). The inconsistent trends between winter and

summer could also be rationalized considering their different source regions, i.e., the PRD region and Northern China during the winter vs. South Asia in the summertime. Overall, the chemical tracer data indicates crop residue burning has been reduced over the decade, perhaps indicating some success in measures such as prohibiting crop burning and crop straw utilization recently implemented in China (Ren et al., 2019). The lack of a consistent declining trend in levoglucosan, on the other hand, implies that biomass burning remains largely uncontrolled and would continue to be a significant PM pollution source.

The three metal species, Pb, Zn and Cu, have been consistently detected in the $PM_{2.5}$ samples over the decade, providing opportunities to probe their associated sources. The three display a strong seasonal contrast, with wintertime concentration levels more than twice those in the summer for Cu and Zn and five times for Pb. The strong seasonality is a characteristic indication for their regional/super-regional origin, in consistent with the cluster analysis results (Figure 5).

Cu and Zn are associated with metal processing industries. Over the decade, the Zn level in the winter has been dropping

steadily, at a rate of -6.6% per year while the wintertime Cu dropped at a rate of -5.8% per year. On the other hand, their summertime change rates were indiscernible from zero (Figure 6 and Table 3). Cumulatively, from 2008 to 2017, approximately 40% reduction was realized for these two metals (Table 2). The significant reductions were likely indicators of benefits from industrial upgrading following the promulgation and implementation of Guangdong "double transfer" policy (industry and labor transfer away from primary industries) since 2009 (Zhong et al., 2013; Yang et al., 2017).

Pb is likely dominated by coal combustion. This source deduction is derived from data collected from a different project, in which we deployed an online XRF spectrometer to monitor hourly concentrations of As, Se, and Pb in Hong Kong from August 2019 to February 2021. The data show strong correlations of Pb with As and Se ($R$>0.80) (Figure S7), two well-known tracers for coal combustion (Tian et al., 2010), providing compelling supporting evidence for coal combustion as a dominant source for Pb. Over the decade, wintertime Pb has displayed a continuous dropping trend at a rate of -7.7% per year, implying

effectiveness in reducing coal combustion emissions in the PRD region and in Northern China. It is also worth noting in the last three years (2015-2017) of the study decade the reduction of the three metals stalled, suggesting that more stringent actions are needed for further reduction in the upcoming years.

Al and Si are classical marker elements for dust particles. They have also decreased over the decade, by -37% for Al and -46% for Si. The two elements are highly correlated, reflecting their common material sources and spatial origins. They display

a distinct seasonality common to the regional sources, i.e., wintertime abundance is notably higher than the summertime. The decreasing rates for wintertime concentrations (-5.3% $yr^{-1}$ for Al and -7.0% $yr^{-1}$ for Si) are more significant than the summertime in terms of both mass concentration and percentage change (Table 3). The decline became flat in 2016-2017





(Section S5), indicating that the current policies started to be less sufficient in reducing the dust contribution.

**3.7 El Niño-Southern Oscillation events**

A closer examination reveals that in the overall monotonic trend component (blue dotted line in Figure 4) of most PM species (PM$_{2.5}$, SO$_4^{2-}$, NO$_3^-$, etc.), 2011 is an anomaly year showing higher concentrations than the preceding and the succeeding years. This resemblance in patterns across the various PM$_{2.5}$ components implies that a macro-factor, for example, sporadic meteorological El Niño/La Niña events, might be at play in influencing the temporal variation.

El Niño-Southern Oscillation (ENSO) events randomly occur during the irregular changes of oceanic temperature among
tropical Pacific Ocean, with El Niño events associated with increase in ocean temperature and La Niña events associated with decrease in ocean temperature. During the events, atmospheric pressure above the Pacific Ocean changes and thus causes the shift of Walker Circulation as well as the distortion of pollutant airflow towards Hong Kong (Yim et al., 2019). The El Niño effect typically leads to a rise in rainfall, less northerly/north-easterly winds, and higher wind speed in Hong Kong (Wang et al., 2019; Yim et al., 2019), thus enhancing the dispersion of regional pollutants. The La Niña effect is associated with opposite
changes in rainfall and wind, thus impeding the dispersion of air pollutants.   Over the decade, there were two El Niño and three La Niña events that lasted for at least 2 months. The strength of ENSO can be classified by the Nino 3.4 index based on the averaged sea surface temperature (SST) anomalies in the Pacific Ocean region. This classification scheme results in five broad groups (Table S6), that is, neutral (0-0.49), weak (0.5-0.99), moderate (1-1.49), strong (1.5-1.99), and very strong (≥ 2) (the numbers in the parentheses indicate the SST anomaly). The rainfall, wind direction, and wind speed at TW under each
level of ENSO events were compared with those on the normal days (i.e., neutral event) and summarized in Figures 9 and S12. The rainfall during all El Niño events was close to that during neutral conditions in Hong Kong but there was a notable reduction of rainfall during strong La Niña events. In terms of wind conditions, more westerly air masses (wind direction > 180°) were transported during very strong El Niño and moderate to strong La Niña events, while the elevated wind speed generally occurred during weak to moderate El Niño events.

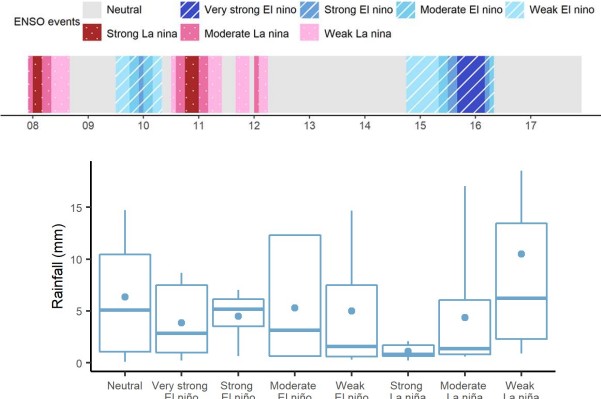


**Figure 9.** The temporal variation in strength of ENSO events in 2008-2017 period (top), and the changes of rainfall (bottom) under different strength levels of ENSO events.

The changes in meteorological conditions were hard to be visualized and quantified. To better investigate the effect of
ENSO in Hong Kong, a multiple linear regression (MLR) was established between the observed concentrations and a list of meteorological variables including temperature (*Temp*), RH, seasonal components, and ENSO events, as shown in Eq. (4). For simplicity, the definitions of seasons here are based on calendar months, with spring corresponding to March-April, summer





May-August, fall September-October, and winter November-February.

$$X_m = \beta_1 Year_m + \delta_1 Season_m + \beta_2 Temp_m + \beta_3 RH + \delta_2 ENSO \qquad (4)$$

where $X_m$ is the monthly averaged time series, $\beta$'s are the coefficients of parametric variables (i.e., Year, Temperature, RH), and $\delta$'s are the coefficients of two dummy variables (i.e., season and ENSO event).

This MLR equation is able to explain additional variance by the ENSO variables (0.63 – 11.7%) without any multicollinearity issue (i.e., generalized variance inflation factor < 5). The prediction from this model is reasonable, producing a slope of ~ 0.8 and $R$ values of 0.58 – 0.86. Briefly, the coefficients for the year ($\beta_1$'s) capture the decline of species and
approximately match the GLS-ARMA results. Seasonal variations were successfully reflected by the coefficients for season ($\delta$'s), temperature ($\beta_2$), or RH ($\beta_3$). For example, positive summer coefficients in V/Ni indicate higher-in-summer, negative spring coefficients in levoglucosan indicate higher-in-winter, negative temperature coefficients in $NO_3^-$/hopanes indicate stronger gas-particle partition or degradation.

The random ENSO events impose different impacts on gaseous and particle pollutants. Significant enhancement of $SO_2$
and $NO_x$ (i.e., $p$-value of $\delta_{2,La\ niña}$< 0.05) was found owing to the La Niña effect while no changes was observed for $O_3$. On the contrary, significant and positive coefficients of the strong La Niña effect were obtained for all $PM_{2.5}$ pollutants except V and Ni. The coefficient was particularly high and positive for some ($SO_4^{2-}$, $NO_3^-$, $NH_4^+$, OC & levoglucosan, $\delta_{2,Strong\ La\ niña}$ from 1.6 to 61) but less for some regional source species (Pb, Cu, $\delta_{2,Strong\ La\ niña}$ from 0.0089 to 0.029). In other words, the concentration of those species was typically high under strong La Niña events in comparison with neutral days. This could be
explained by the significantly suppressed rainfall during the strong La Niña event (Figure 9), where the highly water-soluble ions and levoglucosan were removed to a lesser extent via wet deposition and thus maintained higher concentrations than the normal days. Regardless of the significant level of coefficients, the El Niño effects are generally opposite to the La Niña effect, implying that the enhancement of pollution dispersion/deposition could happen during El Niño events (Table S7).

**Table 4.** Summary of the multiple linear regression of the time series of gaseous and particles pollutants.

| Species | Coefficients[1] | | | | | | |
|---|---|---|---|---|---|---|---|
| | Year | Spring | Summer | Fall | Temp | RH | Strong La Niña |
| **Gaseous pollutants** | | | | | | | |
| CO | **+21*** | *+3.1* | *+21* | *+96* | **-35*** | *+4.6* | *+21* |
| $SO_2$ | **-0.87*** | *+2.1* | *+4.2* | *+0.059* | *-0.2* | *-0.08* | *+4.4** |
| $NO_x$ | **-3.1*** | **+17*** | *+14** | *-0.35* | **-3.3*** | *+0.62** | *+11* |
| $O_3$ | **+1.2*** | *+6.8** | *-3.8* | *+13*** | *+0.36* | **-0.92*** | *-0.69* |
| **Particle pollutants** | | | | | | | |
| $PM_{2.5}$ | **-0.93*** | *+6.5*** | *+1.1* | *+7.3** | **-0.82** | **-0.77*** | **+16*** |
| $SO_4^{2-}$ | *-0.19* | *+2.8*** | *+0.16* | *+3.9*** | *-0.14* | *-0.2*** | *+3.4* |
| $NO_3^-$ | **-0.16*** | *+0.58** | *+0.51* | *+0.24* | **-0.15*** | *-0.021* | *+2.1*** |
| $NH_4^+$ | *-0.06* | *+0.97*** | *+0.39* | *+1.5*** | *-0.15*** | *-0.071*** | *+1.6*** |
| Al | **-10*** | *+50* | *-35* | *+15* | *+3.3* | **-7*** | *+120*** |
| Si | **-19*** | *+96* | *-54* | *+8.5* | *+9.7* | **-17*** | *+330*** |
| V[a] | **-0.71*** | *+16*** | *+14*** | *+3.3* | *-0.051* | *+0.31* | *+5.6* |
| Ni[a] | **-0.29*** | *+4.1*** | *+3.4*** | *+1* | *-0.031* | *+0.057* | *+1.7* |
| Pb | **-2.7*** | *+3.1* | *-4.6* | *+7.7* | *-1.1* | **-2.3*** | *+29*** |
| Zn | *-5.3* | *+29* | *-39* | *+7.9* | *-0.56* | **-6.1*** | *+73* |
| Cu | **-0.8*** | *+4.6* | *-0.47* | *+2.7* | *-0.43* | **-0.71*** | *+8.9** |
| $K^+$ | **-27*** | *-23* | *-82* | *-6.4* | *-11* | **-13*** | *+220*** |
| OC | *-0.079* | *+0.27* | *-0.77* | *+0.088* | *-0.11* | **-0.21*** | *+3.3*** |
| EC | **-0.15*** | *+0.19* | *+0.31* | *-0.025* | *+0.0003* | *-0.0091* | *+0.74*** |
| Hopanes[2] | **-0.045*** | *+0.069* | *+0.012* | *-0.078* | **-0.022** | *+0.0069** | *+0.26*** |
| Levoglucosan[2] | *+0.64* | **-32*** | *-12* | *-4.1* | **-4.6*** | **-1.9*** | *+61*** |

[1]Asterisks denote the coefficient of each variable significantly different from zero: * $p < 0.05$, ** $p < 0.01$, *** $p < 0.001$; coefficients of high significance are marked in bold. [2]The concentration unit of the labelled species is $ng/m^3$ and the unit of the other species is $\mu g/m^3$.





## 4. Conclusions

In this study, we analyzed the 10-year (2008-2017) time series of PM$_{2.5}$, its major components, and select source markers in an urban site in Hong Kong by the STL-GLS-ARMA method. The data set were obtained by following a regular 1-in-6-day
sampling schedule that ensures temporal representativeness and adheres to well-established chemical speciation analysis protocols adopted by the USEPA. In addition, organic molecule maker compounds (i.e., levoglucosan and hopanes) were also measured for this 10-year sample set. Such a long time series of PM$_{2.5}$ chemical composition data derived using a consistent sampling and analysis methodology are rare in China and elsewhere in Asia, thus providing uniquely valuable data to support studies of control measure evaluation and formulation for the region and offering a useful reference for other provinces in
China in evaluating emission control policies.

All PM$_{2.5}$ components were found reduced, with the overall PM$_{2.5}$ mass dropping at -1.5 µg m$^{-3}$ yr$^{-1}$ and by a cumulative rate of 40% (from 37.5 to 22.4 µg m$^{-3}$). The individual contributors to the PM$_{2.5}$ reduction are sulfate (-0.36 µg m$^{-3}$ yr$^{-1}$), OM (-0.25 µg m$^{-3}$ yr$^{-1}$), nitrate and EC (each at -0.17 µg m$^{-3}$ yr$^{-1}$), ammonium (-0.12 µg m$^{-3}$ yr$^{-1}$), and others (-0.39 µg m$^{-3}$ yr$^{-1}$). A disproportional reduction was noted between the precursor gases SO$_2$ (-62%) and NO$_x$ (-36%), and their secondary products
SO$_4^{2-}$ (-40%) and NO$_3^-$ (-66%) because of the complexity in their formation chemistry and formation process spatial scale not confined locally to Hong Kong. A steadily declining trend in EC and hopanes was recorded, achieving a cumulative decrease of 60% and 75%, respectively, in their ambient concentrations. These reductions verify the effectiveness of a series of control measures to reduce vehicular emissions by the Hong Kong government. In comparison, the reduction of OC was much modest, at 23%, which reflects the many more contributing sources as well as important secondary formation contribution to OC.

Two biomass burning tracers, K$^+$ and levoglucosan, displayed strong seasonality in both ambient abundance and 10-year variation trend, as the PRD and Northern China being the source region in the wintertime while South Asia being the source region in the summertime. Wintertime K$^+$ showed a definitive decline trend at a rate of -7.5% yr$^{-1}$ and a cumulative -60% reduction while the decline of wintertime levoglucosan was hardly discernable from zero. In the summertime, neither K$^+$ nor levoglucosan showed a clear decreasing trend. The two tracers track different types of biomass burning, with K$^+$ more
representative of crop residue burning while levoglucosan tracking burning of cellulose. Collectively, the biomass burning tracers indicate that crop straw burning has been reduced over the decade but biomass burning remains a largely uncontrolled regional/super-regional PM$_{2.5}$ sources for Hong Kong.

The 10-year data of Zn, Cu, and Pb showed a cumulative reduction of -40%, -43%, and -60%, respectively. All three metals had strong seasonality, with winter concentrations much higher than the summertime, as metal processing
industries/coal combustion from the GBA region and Northern China as the source regions. Their significant reductions in wintertime (-6.6%, -5.8%, and -7.7% yr$^{-1}$ for Zn, Cu, and Pb, respectively) suggested benefits from measures such as industrial upgrading, coal combustion emission reduction that were implemented over the decade. The reduction for all three metals in the last three years (2015-2017) had stalled, signaling new measures are needed for their further reduction. Dust in Hong Kong's PM$_{2.5}$ mainly comes from regional contribution. The dust components in PM$_{2.5}$ decreased, by-37% for Al and -46%
for Si, over the decade, indicating success in controlling dust generation activities in the region.

Finally, the long-time series reveals that 2011 is an anomaly year in that most PM$_{2.5}$ components were elevated above the adjacent years. By establishing a multiple linear regression model, we show that the heightened strong La Niña events in 2011 resulted in unusually low rainfall, which in turn reduced the removal via wet deposition of aerosol constitutes. In concluding, the long-term chemical speciation data of PM$_{2.5}$ starting as early as 2008 in Hong Kong, one of the important cities in the Great
Bay Area, could be useful for a multitude of purposes related to understanding decadal-scale atmospheric composition change and evaluating significant control policies for the region and the nation.



*Data availability.* Measurement data used in this study are available in the data repository maintained by HKUST https://doi.org/10.14711/dataset/EHHRBZ (Yu et al., 2022).

*Author contribution.* WSC and JZY formulated the overall design of the study. WSC, KFL, and XHHH carried out the chemical
analyses for tracers and key major components and data validation. WSC analyzed the data with contributions from KL, AKHL, and JZY. WSC and JZY prepared the manuscript with contributions from all co-authors.

*Competing interests.* The authors declare that they have no conflict of interest.

*Disclaimer.* The content of this paper does not necessarily reflect the views and policies of the HKSAR Government, nor does mention of trade names or commercial products constitute an endorsement or recommendation of their use.

*Acknowledgements.* We thank Hong Kong Environmental Protection Department (HKEPD) for making part of PM$_{2.5}$ compositional data available for this work. This work is supported by Environment and Conservation Fund (ECF99/2017).

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
