# Peer review of "Measurement Report: Ten-year trend of PM2.5 major components and source tracers from 2008 to 2017 in an urban site of Hong Kong, China"

_Atmospheric Chemistry and Physics, 2022_

## Author Comment (AC1)

**General comments**

The essential information for major sources as well as evaluation and planning of control measures in Hong Kong, China, was proposed based on 10 years of long-term monitoring data of  $PM_{2.5}$ . Overall, it is an interesting study. The test and analysis procedures are reliable and the amount of data are sufficient. The manuscript can be accepted with the following revisions:

**Response:** Thank you for the positive comments. Our response to the comments is given in the following. The response text is marked in blue. References cited in this response document are placed at the end.

**Comments**

1) Chlorine loss of sea salt aerosol was always found in the sampling area, however, particulate chloride were not found in the chemical analysis and long term variation analysis. It was also suggested to analyze the trend variation of Criteria gaseous pollutant data.

**Response:** As the reviewer has pointed out that chlorine loss of sea salt aerosol was common and was also observed in our sampling location. The chloride concentrations were at very low levels at our study site over the years (e.g., as low as  $0.09 \ \mu\text{g/m}^3$  in 2011). The chloride data were lumped into "salt", which is the sum of Na+ and Cl- and shown in the 10-year time series plots of monthly averages and annual averages of PM2.5 chemical composition in Figure 2. Shown in Figure 3, "salt" made up a very minor part (<4% in all years) of PM2.5. We now also add the individual temporal variation plots of Na+ and Cl- in Figure S3.

Due to the low concentration levels, the Cl- data were associated with large measurement uncertainties. One more complicating factor was that nylon filter substrates were used during 2008-2010 while quartz filter substrates were used during 2011-2017 for IC analysis of the ionic constituents. It is known that nylon filters had higher efficiency in retaining chloride while quartz filters suffer negative sampling artifacts due to volatilization of chloride following acidification to HCl by co-existing sulfate (e.g., HSO4-) (Tsai and Perng, 1998; Tsai et al., 2005). The known systematic bias for Cl- measurements between the two sampling media used during the ten-year period, together with the low concentration levels, does not adequately long-term variation analysis.

The following text is added to comment on chloride data, in association with the "salt" component:

Line 243 – 244

"Salt", consisting of  $Na^+$  and  $Cl^-$ , was a very minor part of  $PM_{2.5}$ , accounting for less than <4% in all years."

Regarding the trend variation of criteria gaseous pollutants, we'd like to clarify that they are already illustrated in Figure 4 and summarized in Table 2. Extended discussion about the influence effect of gaseous pollutant variation on the secondary formed products (sulfate and nitrate) was provided in section 3.4. Thus, we did not expand the section other than adding a sentence to convey the use of gaseous pollutants in this study.

Line 152 – 153

"The temporal variations of gaseous pollutants serve as additional data valuable for exploring the effects of changes in precursor gases on the secondary formed PM2.5 constituents (e.g., sulfate and/or nitrate)."

**2) Figure 1: The resolution of Figure 1 needs to be improved.**

**Response**: Thank you for careful checking on the resolution of figures. A higher resolution version of Figure 1 is created and incorporated in the manuscript.

3) Lines 100-113: The authors collected blank samples as a background correction? and how were filter samples stored after collection? More information about sampling and storage should be presented in Ms.

**Response:** Field blanks were collected at a frequency of 10% of a sampler's routine operating frequency. They were used to monitor contamination throughout the process from sampling to analysis and for background correction, if necessary. The field blank filters were stored and analyzed following the same procedures as those of  $PM_{2.5}$  sample filters for quality assurance (QA) purposes.

The following text is added to provide more details about the sampling and storage procedures in our study:

Line 112 - 118

"After the collections, the 47-mm Teflon filters were stored in Petri dishes and the 47-mm quartz filters were stored in the dishes lined with aluminum foil. The dishes were sealed with parafilm. While the 20x25 cm quartz filters were folded in half and stored in aluminum foil. All the filters were packed in a thermal bag for transportation to the laboratory and refrigerated under 4°C before chemical analysis. Field blanks were collected at a frequency of 10% of a sampler's routine operating frequency. They were used to monitor contamination throughout the process from sampling to analysis and for background correction. The field blank filters were stored and analyzed following the same procedures as those of  $PM_{2.5}$  sample filters for quality assurance (QA) purposes."

4) Double "in" were found in line 178. Moreover, please recheck the English by a native Editor **Response:** Thank you for careful reading. Line 178 is revised.

**5) *Line 136: Table 1 should be revised to a trilinear table.**

**Response**: We are unsure what is "trilinear table". We now revise the column arrangement of this table to make it more easily comprehended.

6) Line 230: "For winter PM5, the relative importance of OM increased (up from 31% in 2008 to 40% in 2017)... as well as EC (down from 7.8% to 5.3%)." I can't find any value of 31% in 2008 to 40% in 2017 in the wintertime figure (Figure 3).

**Response:** We double checked the values used for the figures and they were calculated correctly, which means that the number in text are somehow copied wrongly from the results. All the numerical values in the text are now re-examined and any similar issue was now corrected.

The following text is now revised with correct percentage values:

Line 238 – 239:

"For winter  $PM_{2.5}$ , the relative importance of OM increased (up from 35% in 2008 to 39% in 2017) while the relative abundance of nitrate decreased (down from 9.1% to 6.6%), as well as EC (down from 6.1% to 4.4%)."

**7) Line 231: Valid numbers should be uniform.**

**Response**: The numbers in text are now rounded to 2 significant figures. (Please refer to the revised sentence in response#6)

8) Line 233: Do authors have any ideas about the decrease percentage of EC in summer was about 4 times higher than that in winter?

**Response:** The changes in percentage (i.e. the species contribution to  $PM_{2.5}$ ) depend on the concentration variations of both the species and the  $PM_{2.5}$  mass. As shown in the table below, EC levels had little seasonal variations and the reductions over the 10-year period was profound (by 59.5% in winter and by 62.7% in summer). On the other hand, the  $PM_{2.5}$  levels decreased to a much more significant extent in winter (by 43.8%) than in summer (by 26.8%). Therefore, the contribution of EC

| EC, $\mu g/m^3$       | 2008  | 2017 | %Change |
|-----------------------|-------|------|---------|
| Winter                | 3.06  | 1.24 | -59.5%  |
| Summer                | 3.06  | 1.14 | -62.7%  |
| $PM_{2.5}, \mu g/m^3$ | 2008  | 2017 | %Change |
| Winter                | 50.2  | 28.2 | -43.8%  |
| Summer                | 18.3  | 13.4 | -26.8%  |
| EC/PM 2.5  | 2008  | 2017 | %Change |
| Winter                | 6.1%  | 4.4% | -27.9%  |
| Summer                | 16.7% | 8.5% | -49.1%  |

to  $PM_{2.5}$  was more notable in summer (by 49.1%) since the reduction in EC level is more significant than that in the PM mass.

9) Figure 3: The percentage values in the chart use an art font that is unclear and needs to be modified.

**Response:** Thank you for the comment. The art styles are now removed for easier reading to audiences in the figure. Valid rounding in 2 significant figures is also updated in the figure.

10) Line 392: "The concentration levels of shipping emission tracers (V and Ni)... respectively", Is this result from Table 2?

**Response:** Yes, it is the result from Table 2. We now quote the table number in text for better tracking of the results.

The following text is now revised:

Line 407 – 408:

"The concentration levels of shipping emission tracers (V and Ni) were reduced by 34% and 36% by mass concentration and at a rate of -0.60 and -0.29 ng m-3 yr-1 (Table 2), respectively."

11) Line 418: "K+ more representative of burning crop residues high in K+ content vs. levoglucosan representative of burning of cellulose, thus all types of vegetative biomass including hill fires." Grammar errors. The sentence components are unclear and there is no cause-and-effect relationship in this sentence.

**Response**: Thanks for the comment. Here we would like to show the examples where  $K^+$  and levoglucosan can represent two different categories of biomass burning. The sentence is revised as below for improved clarity.

Line 433 – 435:

"Specifically,  $K^+$  is a better marker for emissions from burning crop residues, which are typically enriched in  $K^+$ , while levoglucosan, a thermal pyrolysis product of cellulose, is commonly found in burning of all types of vegetative biomass including hill fires."

12) Line 444: "The two elements are highly correlated, reflecting their common material sources and spatial origins", are the correlation values in the literature or your results?

**Response:** The correlation values was computed based on our results with the correlation matrix of the starting and ending years as shown in Figure 5. The *R*-value of correlation results is added in this sentence as additional information. As characteristic elements of crustal materials, the high correlation of these two elements (Al and Si) are commonly found in ambient PM samples.

The following text is now revised:

Line 461 – 462:

"The two elements are highly correlated (R: 0.54 in 205 and 0.87-0.97 in the other years), reflecting their common material sources and spatial origins."

**13) Line 499: Valid numbers in Table 4 should be uniform.**

**Response:** Table 4 summarizes the coefficients resolved from the multiple linear regression. We'd like to clarify that the numbers are uniform in having the same number of significant figures (i.e., 2). Note that the numbers should not round up to the same decimal place, as the concentrations for different species span a few orders of magnitude.

**References:**

Tsai, C.J. and Perng, S.N., 1998. Artifacts of ionic species for hi-vol PM10 and PM10 dichotomous samplers. *Atmospheric Environment*, *32*(9), pp.1605-1613.

Tsai, C.J., Huang, C.H. and Lu, H.H., 2005. Adsorption capacity of a nylon filter of filter pack system for HCl and HNO3 gases. *Separation Science and Technology*, 39(3), pp.629-643.

---

## Author Comment (AC2)

**General comments**

*The manuscript presents a ten-year data set of PM$_{2.5}$ major components and source-specific tracers at an urban site in Hong Kong, South China. The authors investigate the trends of these compounds and evaluate the influence of emission control on their variations. They also discuss the impact of ENSO events on the abnormal change in PM$_{2.5}$ components, especially in 2011. Overall, I think the research is quite interesting and valuable to the community. I recommend the manuscript to be published in the journal after considering the following specifics:*

**Response:** Thank you for the positive comments. Our response to the comments is given in the following. The response text is marked in blue. References cited in this response document are placed at the end.

**Major concerns:**

1) *There should be emission inventory data in Hong Kong and the PRD. If so, the authors are suggested to compare the long-term trends of PM$_5$ species (tracers) with the variations of local and regional emission inventories. For example, when the authors discuss the long-term variations of SO$_2$ and EC, they list the major emission control measures implemented in Hong Kong (Figure 7 and 8, respectively). Are the changes in these species consistent with the variations in emission inventories in Hong Kong?*

**Response:** We agree with the reviewer that a comparison of the inventory emission data and ambient concentrations would be informative, as one of the data objectives for ambient monitoring is to verify the effectiveness of control measures, thus the broad accuracy of the emission inventories. However, emission is only one of the many factors affecting the ambient concentration of a pollutant. The link between emission and ambient concentration is further complicated by atmospheric transformation processes for secondary pollutant (e.g., sulfate). It is beyond the scope of this study to provide a full examination and comparison between the trends of the emission inventory data and ambient concentrations, which will need emission-based air quality modeling work. Additionally, emission inventories for speciated PM are generally not available. For example, there is no EC Emission inventory available for Hong Kong.

We note that SO$_2$ emission Inventories are available for both Hong Kong and the Guangdong province. As an illustration of comparison between emission inventories and ambient measurements, we now expand the discussion of 10-year trend of sulfate by inclusion of the emission inventory data for SO$_2$. We extracted the long-term emission inventories for Hong Kong and Guangdong from the HKEPD website (https://www.epd.gov.hk/epd/english/environmentinhk/air/data/emission_inve.html) and the MEIC data platform (version 1.3, http://meicmodel.org/?page_id=541&lang=en).

Figure R1 shows the emission inventories of SO$_2$ for Guangdong and Hong Kong. The top two sources for SO$_2$ emissions in Hong Kong are power plants and marine vessels while the major SO$_2$ sources in Guangdong are power plants and industries. The emission and ambient concentration trends, normalized to 2018, are examined in Figure R1(c), showing that the yearly variation of ambient SO$_2$ concentrations at the study site (TW AQMS) was very similar to the total SO$_2$ emission trend from Hong Kong and SO$_2$ emission from power plants in Guangdong.

There was a notable reduction of emissions from power plants in Hong Kong in 2010 owing to the emission caps stipulated by the first Technical Memorandum issued by the HKEPD. A range of marine control measures since 2014 further reduce the SO$_2$ emissions from marine vessels. The regulation toward shipping industry on switching the low-sulfur content fuel for vessels entering Hong Kong helped in reducing extra 18% (-2920 tons) SO$_2$ emissions from navigation sector in 2016 (Figure R2b).

In Guangdong, the drop of SO₂ emissions begin in 2012, mainly due to the reduced contribution of power plant and industrial sources (Figure R2a). Overall, the changes in SO₂ concentrations during the 10-year period are consistent with the SO₂ emissions estimated for the GBA.

[Figure]

**Figure R1 (new Figure S8).** The ten-year changes in percentage share of emissions by sources (columns) and the variations in total SO₂ emissions (solid red line) in (a) Guangdong and (b) Hong Kong, with (c) comparing the ten-year trends of ambient SO₂ at TW and major emission sources of SO₂ in Hong Kong and Guangdong for the period of 2008-2017.

[Figure]

**Figure R2 (new Figure S9).** The changes in the emissions from individual source sectors in (a) Guangdong and (b) Hong Kong.

The following text is added in main manuscript as supportive information linking the trend in ambient SO₂ ambient concentration with the data based on emission inventories:

> Lines 353-361:
> "As a criteria gaseous pollutant, $SO_2$ has been extensively studied and its emission inventories for Hong Kong and Guangdong province are available (HKEPD, 2021b; Li et al., 2017; Zheng, 2018). The $SO_2$ emission inventory data for our study decade are shown in Section S3 in Supplementary information. The top two sources for $SO_2$ emissions in Hong Kong are power plants and marine vessels while the major $SO_2$ sources in Guangdong are power plants and industries (Figure S8). The emission and ambient concentration trends of $SO_2$, normalized to 2018, are examined in Figure S8c, showing that the yearly variation of ambient $SO_2$ concentrations at TW was similar to the total $SO_2$ emission trend from Hong Kong and $SO_2$ emission from power plants in Guangdong. Overall, the changes in ambient $SO_2$ concentrations at TW during the 10-year period are consistent with the $SO_2$ emissions estimated for the GBA."

A new section (Section S3) is added to the SI file, showing Figures R1 and R2 and briefly describing the $SO_2$ emission inventories in Hong Kong and Guangdong.

2) *Recent studies have demonstrated that organic compounds, such as levoglucosan and hopanes, are not stable in the atmosphere as previously thought. As Table 2 shows, the levels of ozone continue to increase at the site. This implies that the atmospheric oxidation capacity is increasing. The authors should add some discussion about the influence of the increase in oxidation capacity on the decrease of organic species.*

**Response:** Our monitoring location (TW AQMS) is in an urban environment. The increasing trend of $O_3$ (+4.6% $yr^{-1}$ in winter and +2.6% $yr^{-1}$ in summer) at this site is mostly accounted for by the attenuated $NO_x$ titration since NOx has continuously been dropping at a rate of ca. -3% $yr^{-1}$. We calculated $O_x$ (= $O_3$ + $NO_2$) at TW site to characterize the atmospheric oxidation capacity (AOC) and found that the AOC over the 10-year period show little discernable change. Nevertheless, we agree with the reviewer that the oxidation capacity of the atmosphere would have impact on the species concentrations. However, the low-resolution nature of our organic tracer data (i.e., daily average), is ill-suited to ascertain the impact of changing atmospheric oxidative capacity on organic degradation rate, and further on concentration levels of the organic markers (e.g., levoglucosan and hopanes).

[Figure]

**Figure R3**. Ten-year variation of $O_x$ (=$O_3$+$NO_2$), shown as the normalized percentage against the level in 2008, at monitoring site in the urban environment of Hong Kong

The effect of oxidation capacity changes on degradation remain unclear from the analysis of measurement data. To keep the paper concise and our focus on reporting the temporal variation, we add one sentence to mention the possible influence of oxidation.

> Line 445-447:
> "We also acknowledge that the ten-year trend in an organic tracer like levoglucosan could be affected by long-term change in atmospheric oxidation capacity, which would exert its impact through atmospheric degradation kinetics."

3) *In section "3. 4 Secondary inorganic aerosol components", the authors discuss the uneven reduction of SO$_2$-sulfate and NOx-nitrate. How about the temporal trends of sulfur oxidation rate (SOR) and nitrogen oxidation rate (NOR)? The changes in SOR and NOR might provide additional information about the formation chemistry of sulfate and nitrate.*

**Response:** Thank you for the suggestions. We calculated the SOR and NOR in winter and summer using Eq. (R1). The 10-year variations of these two ratio quantities are shown in Figure R4.

$$SOR = \frac{nSO_4^{2-}}{nSO_4^{2-} + nSO_2} \;; NOR = \frac{nNO_3^-}{nNO_3^- + nNO_2} \quad where\ n\ refer\ to\ molar\ concentration \quad (R1)$$

The SOR value was higher than 0.1 for our data in all years, indicating the significant oxidation and in

line with the large regional transport contribution for sulfate. In the transformation of $SO_2$ to sulfate, multiple oxidants could be at work (e.g, gaseous OH, $H_2O_2(aq)$, $O_3(aq)$, etc) (Xue et al., 2019). The aqueous oxidation mechanisms are likely not closely coupled with gaseous oxidation capacity. Additionally, the aqueous pathways are dependent on cloud availability. As shown in Figure R5, no clear co-variation temporal patterns can be discerned for ozone and sulfate. This result, although crude, reflects that a straightforward linkage is absent between one of the gaseous oxidants (i.e., O3) and sulfate as an oxidation product of $SO_2$.

The NOR value was typically low (< 0.05) (Figure R4). Note that $PM_{2.5}$ nitrate is only one part of the oxidation products of NOx. Other forms of nitrate, such as gaseous $HNO_3$, nitrate on coarse particles, organic nitrates, etc., were not reflected in the NOR calculation, but they could be comparable in abundance to PM2.5 nitrate. Additionally, $PM_{2.5}$ nitrate (mainly in the form of ammonium nitrate) is semi-volatile and its partition in the particle phase is strongly affected by temperature and relative humidity. For example, the stack difference NOR between summer and winter is more likely driven by temperature. As such, we feel using NOR to indicate formation pathway is not well-grounded.

In summary, the complex oxidation chemistry of $SO_2$ to sulfate, and the multiple significant forms of nitrate plus the semivolatile nature of ammonium nitrate make the SOR and NOR quantities not readily indicative for formation chemistry insights. Thus, we decide not to explore these two ratios in our manuscript.

[Figure]

**Figure R4.** The seasonal variation of meteorological condition (wind speed and wind direction), ozone, SOR and NOR (from top to bottom).

**Minor comments:**
1) *Page 8 line 233-234. The authors state that the decrease in EC is due to local control of vehicular emissions. However, in addition to vehicular emissions, EC could be emitted from biomass burning and shipping exhaust. In fact, the tracers of the latter two sources (e.g., $K^+$ and Ni) also continued to decrease (Table 2). The "%Relative change" of EC is close to that of $K^+$.*

**Response:** Thank you for the question. Generally speaking, the review is correct that combustion sources, such as vehicular emissions, biomass burning, and ship emissions, all contribute to EC. For our study location, these three sources mentioned by the reviewer have distinct seasonality. Vehicular and shipping emissions are mainly of local origin, thus showing little seasonality. Biomass burning is largely regional/super-regional, displaying clear seasonality of higher in the winter and lower in the summer.

Here, we use season-specific Sen's slope to compare the seasonality and thus verify the origin of emission sources with clustering results of the correlation matrix. The closer the season-specific Sen's slope between winter and summer, the more contribution of local sources to the species. EC shows a comparable Sen's slope of -0.19 $\mu g\ m^{-3}\ yr^{-1}$ in winter and -0.18 $\mu g\ m^{-3}\ yr^{-1}$ in summer, indicating the strong local source contribution in EC. Ni also has similar Sen's slope values in winter and in summer (-0.37 and -0.42 $ng\ m^{-3}\ yr^{-1}$). On the other hand, $K^+$ has a much higher Sen's slope in winter than in summer (-62 vs. -11 $ng\ m^{-3}\ yr^{-1}$). Although the %relative change of EC is close to that of $K^+$, their overall temporal and seasonal variation were quite different from each other. In addition, EC had moderate to good correlations (*R*-values: 0.41-0.74) with species which are emitted/generated locally such as hopanes (tracer for vehicular emissions) and $NO_x$ but was poorly correlated with $K^+$ (*R*-value: 0.19). Our source apportionment study for data in 2015 shows that ship emissions make negligible contributions to EC at TW (See Table S4b in Chow et al., 2022). In conclusion, the relevant monitoring data strongly indicates EC at TW was mainly from local vehicular emissions and its reductions over the years were in excellent correspondence to local vehicular emission control measures.

2) *Page 12 Figure 5. SO₂ had a local source in 2008 but a regional source in 2017. What is the explanation?*

**Response:** Thanks for your comment. We computed the spearman correlation between species in each year to examine and to preliminary classify the sources into local and regional groups by hierarchical clustering. The correlation of $SO_2$ with species of local origins such as EC, V and Ni became relatively poor in 2017 whilst the correlation with regional species such as Zn and $K^+$ increased in the same year (Figure R7). We believe it is because the local $SO_2$ emissions have been greatly reduced since 2010, when the emission caps for power plants in Hong Kong came into effect, so that the regional transport of $SO_2$ began to play a relatively more important role.

In addition, the annual percentage changes of ambient $SO_2$ was similar to those from local emission inventory in the beginning years (2008-2010) (ambient: -45%, inventory: -49%), but was lower than local emission inventories in the ending year (2008-2017) (ambient:-62%, inventory:-77%). This further supports that there is a shift of dominant emissions source of $SO_2$ from local to regional/super-regional over the decade. The below text is added to comment on the shifting of $SO_2$ dominant sources over the decade.

> Lines 313-315
> "It is of interest to note that $SO_2$ shifted from the local cluster to the regional cluster over the decade, reflecting the changing relative importance of local vs. regional emission sources of $SO_2$ over the years (see Section S3 for more details)."

3) *Page 19 Table 4. The coefficients in WINTER are not listed in Table 4.*

**Response:** Table 4 lists the regression coefficients for the variables in the multiple linear regression Eq. (4).

$$X_m = \beta_1\ Year_m + \delta_1 Season_m + \beta_2\ Temp_m + \beta_3 RH + \delta_2 ENSO \qquad (4)$$

In Equation (4), there are three parametric variables (Year, Temperature, and RH) and two categorical (or dummy) variables (Season and ENSO event). The coefficients for the categorical variables are derived by selecting one of the categories as the reference (i.e., winter in our case) and calculating the

relative values of each variable with respective to the reference. Hence there is no coefficient for winter in the table. This explanation is now added as part of the table footnote. As Table 4 data is portion of Table S7, we now deleted Table 4 from the main text (the explanation for no coefficient for winter is added as Table S7 footnote). Related to Table 4 data, we also create a new Figure (Figure S15), visually showing the varying magnitude of the effect of Strong La niña event on various gaseous and particle species.

References:
Chow, W. S., Huang, X. H. H., Leung, K. F., Huang, L., Wu, X., and Yu, J. Z., 2022. Molecular and elemental marker-based source apportionment of fine particulate matter at six sites in Hong Kong, China. *Sci. Total Environ.*, 813, 152652.

Xue, J., Yu, X., Yuan, Z., Griffith, S.M., Lau, A.K., Seinfeld, J.H. and Yu, J.Z., 2019. Efficient control of atmospheric sulfate production based on three formation regimes. *Nature Geoscience*, *12*, 977-982.

---

## Author Response (AR2)

Manuscript ID: acp-2022-100
Title: "Measurement Report: Ten-year trend of PM2.5 major components and source tracers from 2008 to 2017 in an urban site of Hong Kong, China"
Journal: Atmospheric Chemistry and Physics
Author(s): Wing Sze Chow, Kezheng Liao, X. H. Hilda Huang, Ka Fung Leung, Alexis K. H. Lau, Jian Zhen Yu

**Comments to the author**:

The authors have reasonably addressed the comments of the two anonymous referees and they have modified their manuscript accordingly. However, many alterations and corrections are needed for both the Main text and Supplement before the manuscript can be published in ACP:

**Authors' Response:**

We thank the editor for the careful reading and suggestions. We have taken the suggestions and made the revisions accordingly in the revised manuscript and the supporting document. Note we did not provide a point-by-point response, as all the suggested revisions have been made without any exceptions.

**Detailed comments from the editor**

Main text:

Line 16: Replace "displayed significant" by "displayed a significant".

Line 37: GBA was already defined in line 34; it should not be defined again; therefore, replace "Area (GBA)" by "Area".

Line 40: Replace "in term" by "in terms".

Line 94: Abbreviations and acronyms should be defined (written full-out) when first used; therefore, replace "LOESS (STL)" by "locally estimated scatter plot smoothing (LOESS)"; note that STL is defined in line 155.

Line 125: OC and EC were already defined in line 50; they should not be defined again; therefore, replace "Organic carbon (OC) and elemental carbon (EC)" by either "Organic carbon and elemental carbon" or "OC and EC".

Line 130: Replace "al.,2008" by "al., 2008".

Line 148: HKEPD was already defined in line 72; it should not be defined again; therefore, replace "Hong Kong Environmental Department (HKEPD)" by either "Hong Kong Environmental Department" or "HKEPD".

Lines 150 and 151: Abbreviations and acronyms, here "HKUST ENVF", should be defined (written full-out) when first used.

Line 154: Replace "in supplementary" by "in the supplementary".

Line 155: Replace "with LOESS" by "with the LOESS".

Line 157: STL was already defined in line 155; it should not be defined again; therefore, replace "decomposition (STL)" by "decomposition".

Line 162: LOESS was already defined in line 94; it should not be defined again; therefore, replace "locally weighted regression (LOESS)" by "LOESS".

Line 169: Replace "of interquartile" by "of the interquartile".

Line 174: Replace "from STL" by "from the STL".

Line 177: Replace "of changing" by "of the changing".

Line 182: It should be indicated what the symbols φ and θ denote.

Line 187: Replace "indictors" by "indicators".

Line 189: Replace "in supporting" by "in the supporting".

Line 192: There is an inconsistency between "Wilcox, 2017" here and "Wilcox, 2018" in the Reference list.

Line 193: Replace "run Mann-Kendall" by "run a Mann-Kendall".

Line 200: Replace "of CAGR" by "of the CAGR".

Line 201: Replace "of supporting" by "of the supporting".

Line 211: Replace "summertime and bring" by "summertime bring".

Line 218: Replace "variations, therefore" by "variations and are therefore".

Line 224: Replace "in summer" by "in the summer".

Line 225: Replace ", however" by "; however".

Line 227: Replace "of 10-year" by "of the 10-year".

Line 232: Replace "computed to be" by "computed as".

Line 242: Replace "than <4%" by "than 4%".

Line 248: Replace "examined annual" by "examined the annual".

Line 255: Replace "degree of freedom" by "degrees of freedom".

Line 256: OLS was already defined in line 177; it should not be defined again; therefore, replace "squares (OLS)" by "squares".

Line 268: Replace "of STL-GLS-ARMA" by "of the STL-GLS-ARMA".

Line 279: Replace "by GLS-ARMA" by "by the GLS-ARMA".

Line 280: Replace "flatten variation" by "flattened variation".

Line 282: Replace "between two" by "between the two".

Line 289: Replace "confine trend" by "confine the trend".

Page 11, last but one line: Replace "in the bracket denote" by "in parentheses denote".

Lines 317 and 333: Replace "select components" by "selected components".

Line 323: Replace "align with" by "alignment with".

Line 324: Replace "displays comparable" by "displays a comparable".

Line 342: Replace "namely, sulfate" by "namely sulfate".

Line 343: Replace "of PM2.5" by "of the PM2.5".

Line 354: There is an inconsistency between "Zheng, 2018" here and "Zheng et al., 2018" in the Reference list.

Line 354: Replace "in Supplementary" by "in the Supplementary".

Line 389: Replace "for higher" by "for a higher".

Line 394: Replace "project, thus" by "project and are therefore".

Line 400: Replace "hese significant" by "These significant".

Line 408: Replace "in Shing" by "in the Shing".

Line 417: Replace "in the top plot" by "in the plot".

Line 422: Replace "show that" by "shows that".

Line 456: There is an inconsistency between "Yang et al., 2017" here and "Yang and Zhu, 2017" in the Reference list.

Line 466: Replace "in 205" by "in 2015".

Line 478: Replace "tropical Pacific" by "the tropical Pacific".

Line 480: Replace "of Walker" by "of the Walker".

Line 495: Replace "in 2008-2017" by "in the 2008-2017".

Line 508: Replace "$\delta$'s" by "$\delta_1$'s" with the "1" in subscript.

Line 513: Replace "was observed" by "were observed".

Line 524: Replace "select source" by "selected source".

Line 525: Replace "were obtained" by "was obtained".

Line 548: Replace "sources for" by "source for".

Line 551: Replace "China as the" by "China are the".
Line 559: Replace "aerosol constitutes" by "aerosol constituents".
Line 569: The abbreviation "HKSAR" should be replaced by what it stands for.
Line 571: Replace "thank Hong" by "thank the Hong".
Lines 578-584: "Chen, Z. et al., 2019" should come after "Chen, W. et al., 2021".
Lines 684-687: "World Health Organization" should come after "Wilcox".

Supplement:
Lines 8 and 46: Replace "The meteorological" by "Meteorological".
Line 20: Replace "The changes in percentage share of emissions by sources and the variations" by "Ten-year changes in percentage share of emissions by sources and variations".
Lines 22, 40, 125, and 254: Replace "The changes" by "Changes".
Lines 23 and 127: Replace "Loess Method and the Generalized" by "Loess Method (STL) and Generalized".
Lines 34 and 238: Replace "The overall" by "Overall".
Lines 36 and 245: Replace "all the pairwise" by "all pairwise".
Lines 37 and 247: Replace "ENSO" by "El Niño-Southern Oscillation (ENSO)".
Lines 44 and 263: Replace "the PRD" by "the Pearl River Delta (PRD)".
Line 47: Replace "are summarized" by "is summarized".
Line 49: Replace "in summer" by "in the summer".
Lines 51, 56 and 58: Replace "in Shing" by "in the Shing".
Line 64: Replace "then calculated" by "then we calculated".
Line 65: Replace "in normal distribution (e.g. the" by "in the normal distribution (e.g., the".
Line 66: Replace "verse theoretical" by "versus the theoretical".
Line 67: Replace "in Q-Q" by "in the Q-Q".
Figure S3: The units in the ordinates should be indicated; furthermore, it is unclear what the # at the end of the legend in the ordinate denotes.
Lines 75 and 82: Replace "while that of the remaining is" by "while those of the remaining substances are in".
Figure S5: It is unclear what the # at the end of the legend of the ordinate in Figure S5(b) denotes.
Figure S6: It should be indicated in which units the concentrations were.
Line 103: The "2" of "SO2" should be in subscript.
Line 107: Replace "begun" by "began".
Line 112: Replace "shows that" by "show that".
Line 113: Replace "TW" by "Tsuen Wan (TW)".
Line 115: Replace "GBA" by "Greater Bay Area".
Line 120: Replace "The ten-year changes in percentage share of emissions by sources (columns) and the variations" by "Ten-year changes in percentage share of emissions by sources (columns) and variations".
Line 122: Replace "Hong Kong and Guangdong" by "Hong Kong (HK) and Guangdong (GD), HK El indicates Public Electricity Generation in Hong Kong".
Figure S9: It should be specified what the small boxes and numbers between the bars denote.
Line 138: Replace "with initially" by "with an initially".
Line 139: I presume that the "s" of "ns" should be in subscript.
Line 139: Replace "Jan is" by "January is".
Line 141: Replace "in inner" by "in the inner".
Line 153: Replace "robustness weights" by "robustness weight" and replace "respected to" by "respect to".

Line 161: Replace "square with ARMA" by "squares with the ARMA".

Line 162: Replace "of time" by "of the time" and replace "it assumes" by "it is assumed".

Line 165: Replace "Prior from directly running the model, autocorrelation issue on time series by Auto" by "Prior to directly running the model, the autocorrelation issue on the time series by the Auto".

Line 166: Replace "Figure" by "Figures".

Line 177: Replace "select by minimum AIC/AICc were" by "selected by minimum AIC/AICc was".

Line 179: Replace "were not" by "was not".

Line 192: Replace "while the rest is" by "while for the other species it is".

Line 194: Replace "denote for the significance of slope differ" by "denotes that the significance of the slope differs".

Line 197: Replace "by STL method, OLS instead of GLS-ARMA" by "by the STL method, OLS instead of the GLS-ARMA".

Line 199: Replace "from OLS" by "from the OLS".

Line 202: Replace "incorrected in" by "incorrect in".

Line 203: Replace "Concentration estimated" by "Concentrations estimated".

Line 206: Replace "were dominated" by "was dominated".

Line 208: Replace "were generally" by "was generally".

Figure S12: The units in the ordinates should be indicated.

Table S3: The units of the Concentrations should be specified.

Line 224: Replace "with STL-GLS-ARMA" by "with the STL-GLS-ARMA".

Line 227: Replace "of species" by "of the species".

Line 228: Replace "observed in" by "observed in the".

Line 229: Replace "flatten in later year" by "flattened in later years".

Line 231: Replace "started from" by "starting from".

Line 236: Replace "in bracket" by "in parentheses".

Line 240: Replace "summarize the" by "summarizes the".

Line 249: Replace "MLR" by "multiple linear regression (MLR)".

Line 260: Replace "RH" by "relative humidity (RH)".

Page 25, line 3 from bottom: Replace "labeled species" by "the labeled species" and replace "unit of the rest" by "the unit of the other".

Page 25, last line: Replace "of other" by "of the other".

Page 26, middle column: Replace "60 mins" by "60 min".

Page 27, last column: Replace "e.g." by "e.g.,".

---

## Author Response (AR3)

Manuscript ID: acp-2022-100
Title: "Measurement Report: Ten-year trend of PM2.5 major components and source tracers from 2008 to 2017 in an urban site of Hong Kong, China"
Journal: Atmospheric Chemistry and Physics
Author(s): Wing Sze Chow, Kezheng Liao, X. H. Hilda Huang, Ka Fung Leung, Alexis K. H. Lau, Jian Zhen Yu

**Comments to the author**:

**Comments to the author**:

A few corrections are still needed in the Supplement before the manuscript can be published in ACP:

Line 67: Replace "verse the theoretical" by "versus the theoretical".

Line 135: Replace "and the Generalized" by "and Generalized".

Line 169: Replace "with ARMA" by "with the ARMA".

Lines 225-226: Replace "Note that residual unit of hopanes and levoglucosan are ng/m3, while those of the" by "Note that the residual unit for hopanes and levoglucosan is ng/m3, while that for the".

Table S3, in the unit column for Si and V: Replace "µgC" by "µg".

**Authors' Response:**

We thank the editor for the careful reading. We have taken the suggestions and made the revisions accordingly in the revised supporting document. Note we did not provide a point-by-point response, as all the suggested revisions have been made without any exceptions. We also made some additional edits to improve clarity or to correct grammatical errors. These edits are also tracked and marked in blue.

We have uploaded a version of the revised supporting document with all the text changes marked in blue.